# Genomic insights into the 2022–2023 *Vibrio cholerae* outbreak in Malawi

Chrispin Chaguza [1,2,3,4,5,19] ✉, Innocent Chibwe [6,19], David Chaima[7], Patrick Musicha [5,8], Latif Ndeketa[3,8], Watipaso Kasambara[6], Chimwemwe Mhango[8], Upendo L. Mseka[8], Joseph Bitilinyu-Bangoh[6], Bernard Mvula[6], Wakisa Kipandula[9], Patrick Bonongwe[10], Richard J. Munthali [11], Selemani Ngwira[6], Chikondi A. Mwendera[3], Akuzike Kalizang'oma[4,8], Kondwani C. Jambo [8,12], Dzinkambani Kambalame[6], Arox W. Kamng'ona [13], A. Duncan Steele[14], Annie Chauma-Mwale[6], Daniel Hungerford [3,15], Matthew Kagoli[6], Martin M. Nyaga [16], Queen Dube[17], Neil French[3,8], Chisomo L. Msefula[7,20], Nigel A. Cunliffe[3,15,18,20] & Khuzwayo C. Jere [3,8,9,15,18,20] ✉

Malawi experienced its deadliest *Vibrio cholerae* (*Vc*) outbreak following devastating cyclones, with >58,000 cases and >1700 deaths reported between March 2022 and May 2023. Here, we use population genomics to investigate the attributes and origin of the Malawi 2022–2023 *Vc* outbreak isolates. Our results demonstrate the predominance of ST69 clone, also known as the seventh cholera pandemic El Tor (7PET) lineage, expressing O1 Ogawa (~ 80%) serotype followed by Inaba (~ 16%) and sporadic non-O1/non-7PET serogroups (~ 4%). Phylogenetic reconstruction revealed that the Malawi outbreak strains correspond to a recent importation from Asia into Africa (sublineage AFR15). These isolates harboured known antimicrobial resistance and virulence elements, notably the ICE^GEN/ICEVchHai1/ICEVchind5 SXT/R391-like integrative conjugative elements and a CTXφ prophage with the *ctxB7* genotype compared to historical Malawian *Vc* isolates. These data suggest that the devastating cyclones coupled with the recent importation of 7PET serogroup O1 strains, may explain the magnitude of the 2022–2023 cholera outbreak in Malawi.

*V ibrio cholerae* (*Vc*) is a Gram-negative curved-rod-shaped bacterium that causes outbreaks and epidemics of life-threatening severe acute watery diarrhoeal illness, cholera, which is associated with high morbidity and mortality if untreated[1]. Cholera causes ~ 3 million cases globally per year, leading to nearly 100,000 deaths, with a two-fold higher case fatality rate in Africa[2,3]. The World Health Organisation (WHO) recommends the use of highly effective oral rehydration therapy using polymer-based or glucose-based rehydration solutions for mild or moderate infections and intravenous rehydration therapy complemented with antibiotics to treat severe cholera infection[4,5].

*V. cholerae* is primarily transmitted from person to person through faecal contamination routes or poor food hygiene, and from the environment to person via *Vc*-contaminated water reservoirs[6]. Introductions of *Vc* strains from other countries[7,8], during humanitarian crises (war)[6,9] and natural disasters (earthquakes and cyclones)[10,11], which disrupt water and sanitation systems or displace populations towards inadequate and overcrowded living conditions, increase the risk of cholera transmission[1].

More than 200 *Vc* serogroups have been characterised and are differentiated serologically based on the O-antigen of its cell surface

lipopolysaccharide (LPS)[1]. However, only the O1 and O139 serogroups are typically associated with cholera outbreaks and epidemics, particularly in endemic settings[12]. The O1 serogroup is further divided into phenotypically distinct biotypes, namely, classical and El Tor, that evolved from independent lineages and the former is associated with earlier pandemics[12]. The O1 serogroup biotypes are further subdivided into the Ogawa and Inaba serotypes.

Although the modern history of cholera dates to 1817, several accounts of cholera-like illnesses were reported in the ancient times of Hippocrates circa, 300–500 BC[13]. Serogroup O1 was responsible for the early cholera outbreaks associated with the seven cholera pandemics starting from 1817[12]. However, O139 *Vc* strains resembling serogroup O1 strains emerged in the early 1990s, first reported in India[14] and caused outbreaks in Bangladesh in the early 1990s[15,16]. The current seventh cholera pandemic El Tor (7PET) lineage backbone, principally associated with the O1 serogroup and rarely with O139, dates to ~1961 in Sulawesi, Indonesia. E1 Tor subsequently spread globally[8], including the first introductions into Africa around ~1970, and it has persisted ever since[7,17–21]. Detailed phylogeographic analysis revealed fifteen independent introductions of *Vc* into Africa from other continents (designated T1-T15) due to antibiotic-susceptible and multidrug-resistant (MDR) *Vc* lineages from 1970s[7,17,22,23]. In addition to the serogroup and biotypes, the presence and absence of critical virulence factors are widely used to distinguish *Vc* strains. These *Vc* virulence factors include the cholera toxin (CT) carried on the filamentous lysogenic CTXφ prophage[24,25], encoded by *ctxA* and *ctxB* genes. This toxin is responsible for the manifestation of severe watery "rice-water" diarrhoea with ongoing purging in cholera patients[6,26]. The second most important *Vc* virulence factor is the toxin-coregulated pilus (TCP), which is a receptor for the CTXφ phage[27,28], and is encoded by the TCP operon in the Vibrio pathogenicity island (VPI-1)[28]. This pilus is required for *Vc* colonisation of the small intestinal epithelium[29]. These virulence factors are differentially expressed between classical and El Tor biotypes, and others encoded on prophages and pathogenicity islands, including the Vibrio seventh pandemic island II (VSP-II) element, are more commonly associated with El Tor biotypes responsible for the 7th cholera pandemic[30]. In addition, the acquisition of mobile genetic elements (MGE), including plasmids, transposons, integrons and integrative conjugative elements (ICE)[31], provides further context for characterising *Vc* isolates globally. These ICEs include the SXT/R391 family, which was first identified in an O139 isolate in 1993 in India and carried genes conferring antimicrobial resistance (AMR) to sulfamethoxazole/trimethoprim (SXT)[32], and other variants of this ICE have been reported in recent years[33,34].

Malawi experienced its largest cholera outbreak from 2022–2023, with >58,000 cases and >1700 deaths reported countrywide across all 29 districts[11,35]. Besides the devastation caused in Malawi by tropical cyclones Ana and Gombe in early 2022 and Freddy in 2023, specific attributes of the *Vc* strains that may have contributed to the high incidence and mortality of the 2022–2023 cholera outbreak remain unknown. Here, we describe the epidemiology of cases and deaths attributed to cholera during the 2022–2023 outbreak in Malawi and examine the origin and genomic attributes of *Vc* isolates collected as part of the national public health response. Unlike the traditional typing methods widely used to characterise *Vc*, such as serotyping, phage typing, and determination of antibiograms, whole-genome sequencing provides greater resolution, allowing for adequate characterisation of strains and the genetic repertoire of the isolates[12]. We performed whole-genome sequencing of *Vc* isolates collected nationwide and compared them to the contextual historical *Vc* isolates from Malawi and globally. We determined the serogroups, serotypes and biotypes, and genetic similarities, together with the distribution of virulence, and AMR gene profiles of the *Vc* isolates, to understand potential pathogenic traits that may have contributed to the magnitude of the 2022–2023 outbreak in Malawi. Our findings provide the

first insights into the evolution and genomic diversity of the 2022–2023 cholera outbreak-associated *Vc* isolates in Malawi to inform public health strategies to prevent and control current and future cholera epidemics.

## Results

### The 2022–2023 *V. cholerae* outbreak is the deadliest recorded in Malawi

The WHO defines Malawi as a cholera-endemic country with annual outbreaks occurring during the rainy season from November to May. Based on data from the Malawi Ministry of Health (MoH) (Accessed on May 30, 2024; https://cholera.health.gov.mw/surveillance), the 2022–2023 cholera outbreak has to date resulted in 59,156 cases and 1771 deaths in Malawi (Fig. 1a, b). These case and death counts make the 2022–2023 cholera outbreak the largest and deadliest cholera outbreak recorded in Malawi[36]. Based on these estimates, the 2022–2023 cholera outbreak in Malawi has a case fatality ratio (CFR) of 3.0% (1759 of 58,730 cases; 95% confidence interval [CI]: 2.88 to 6%) (Fig. 1c). This CFR is similar to that reported during the 1998–1999 outbreak in Malawi (3.4%; 860 of 25,000 cases) but slightly higher than those observed during 2001–2002 (2.3%; 968 of 33,546) and 2008–2009 (2.17%; 125 of 5751) outbreaks[37]. Similarly, the CFR for the current Malawi outbreak is higher than those reported from Malawi's neighbouring countries; for instance, the CFR was 1.8% (98 of 5414 cases) for the 2017–2018 outbreaks in Zambia and 0.7% (37, 5 of 237 cases) during the 2022–2023 outbreak in Mozambique[38]. On a global scale, the overall CFR for the 2022–2023 outbreak in Malawi is higher compared to the cholera outbreaks that occurred in 2016–2017 in Yemen (0.22%; 2385 of 1103,683 cases)[9] and the 2010–2011 epidemic in Haiti (2.3%; 654 of 29,295 cases)[39] cholera epidemics, but similar to that reported during the 2022–2023 outbreak in Port-au-Prince, Haiti (3.0%; 144 of >20,000 cases)[40]. The CFR estimates for the current outbreak in Malawi varied greatly by district, ranging from 0.62% (3 of 486 cases) to 6.67% (6 of 90 cases) in Mzimba and Kasungu districts, respectively. Similarly, the number of cases ranged from 68 in Ntchisi to 12,683 in Lilongwe (Fig. 1e, f; Table 1). To account for the differences in the population size in each district, we calculated the overall cumulative incidence of cases per district per 100,000 people. The mean incidence was 352.91 (range: 10.68 to 1404) cases per 100,000 people while the incidence of deaths was 8.66 (range: 0.32 to 21.24) deaths per 100,000 people (Table 1 and Supplementary Fig. 1). These data highlight areas to be given the highest priority for public health outbreak prevention and control measures, including oral cholera vaccination[41,42].

Uniquely, the 2022–2023 cholera outbreak in Malawi started towards the end of a typical seasonal cholera outbreak in Malawi, with the first *Vc* cases observed in March 2022, which persisted throughout the dry season until the 2022–2023 rainy season (Fig. 1a). The number of cholera cases and deaths started to increase immediately following tropical cyclone Gombe, one of the largest cyclones recorded in Malawi, which occurred between March 8–14, 2022 (Fig. 1b, Supplementary Fig. 2). The number of cases increased steadily every month from March 2022 as new districts registered cases, leading to a substantial increase of cases along the lakeshore districts before a larger outbreak towards the start of the 2022–2023 rainy season. The larger outbreak peaked in February 2023, before the occurrence of cyclone Freddy in March 2023 (Fig. 1a, c). Whereas past seasonal cholera outbreaks in Malawi mostly occurred in a few districts, especially Machinga, Zomba, Thyolo, Nsanje, Chikwawa, and Phalombe, which are typically at the highest risk during the rainy season[43], the 2022–2023 outbreak spread to all twenty-nine districts (Fig. 1e, f). The cholera outbreaks that occurred in several districts did not entirely overlap with each other; instead, there was a sequential occurrence of outbreaks, starting in the Nsanje district, immediately after cyclone Gombe (Supplementary Fig. 2). Following this initial rise of cases in this

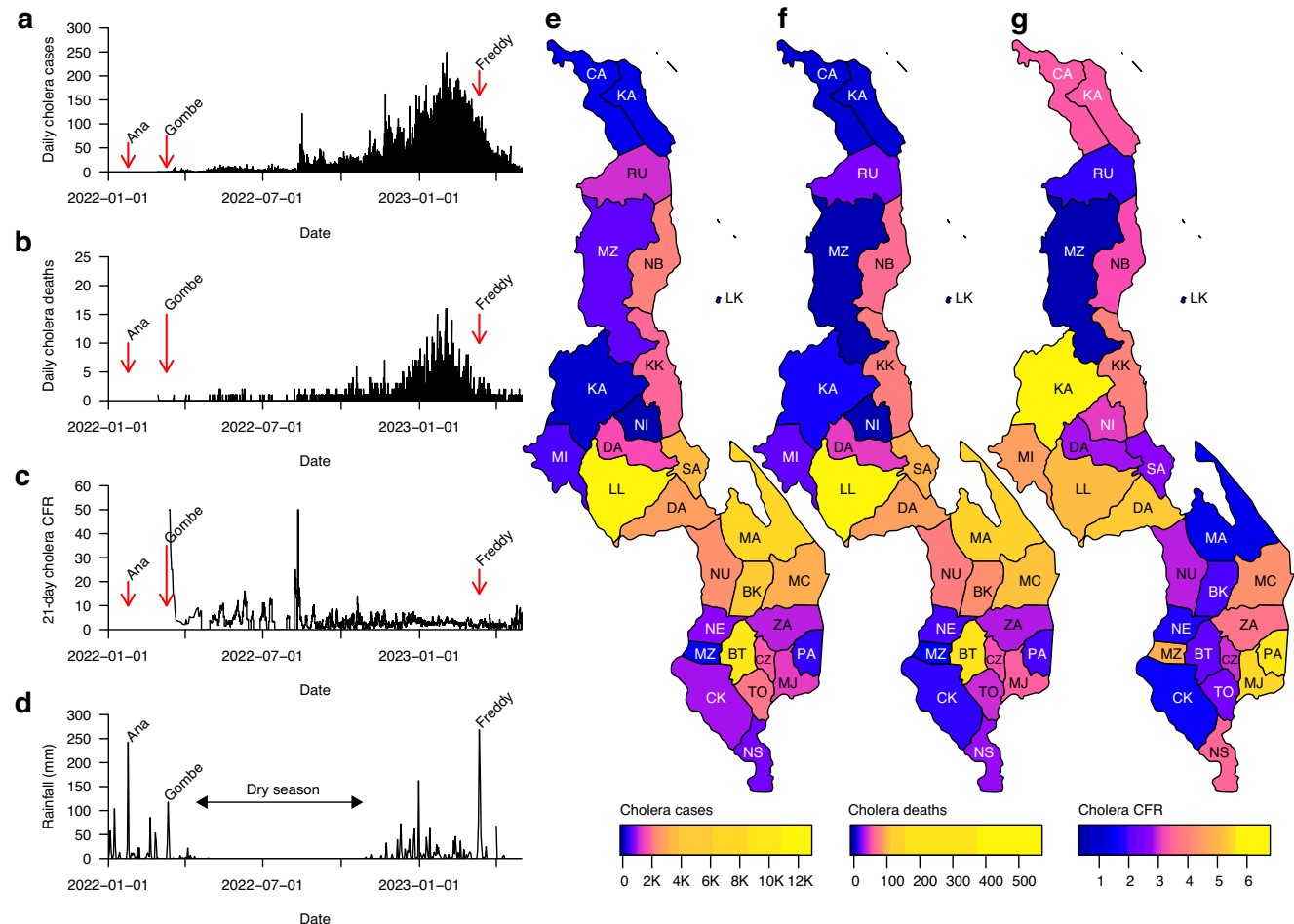

**Fig. 1 | Cases, deaths, and case fatality ratio during the 2022–2023 cholera outbreak in Malawi (data from January 2022 to May 2023). a** Total daily cholera cases. **b** Total daily cholera deaths. **c** Overall cholera case fatality ratio (CFR) based on a 21-day sliding window. The 21-day sliding window was chosen to obtain stable estimates of the CFR, especially during weeks and months with few reported cholera cases. **d** Daily rainfall in millimeter (mm) units obtained from Chichiri weather station in Blantyre, Malawi. **e** Map of Malawi showing the number of cholera cases per district. **f** Map of Malawi showing the number of cholera deaths per district. **g** Map of Malawi showing the CFR of cholera cases per district. Data were obtained from the Public Health Institute of Malawi (PHIM), Malawi Ministry of Health (MoH) data on May 20, 2023 (https://cholera.health.gov.mw/surveillance).

flood-prone district, an increase of cholera cases occurred a few weeks later in nearby districts in the lower Shire region in southern Malawi, including Chikwawa and Neno, which spilled over to Blantyre possibly igniting the spread of cases throughout the country. Cholera cases were reported in Nsanje, Chikwawa, Neno, and Blantyre throughout 2022, including the dry season, leading to the larger outbreaks observed in all districts after the start of the 2022–2023 rainy season (Supplementary Fig. 2). Together, these findings demonstrate the countrywide spread of *Vc*, with variable incidence and temporal spread across districts in Malawi.

### Potential to incorrectly attribute cholera-like cases to *V. cholerae* in Malawi

Studies in Africa and elsewhere have suggested that non-*Vc* bacterial species may cause simultaneous cholera-like infections during cholera outbreaks[44–48]. However, it is currently unknown what proportion of cholera-like cases may be attributed to non-*Vc* bacterial pathogens. To begin to understand this in Malawi, we assessed the proportion of confirmed *Vc* and other bacteria based on whole-genome sequencing (WGS) data generated from presumptive *Vc* isolates from patients presenting with cholera or cholera-like symptoms. All the cultures that were subjected to WGS had yellow colonies on thiosulphate citrate bile salt sucrose (TCBS) agar, suggestive of *Vc* (see methods). We recovered

sufficient genomic data from 68 out of 75 suspected *Vc* isolates, of which 49 of 68 cases (~72%) of the sequenced genomes were confirmed to contain *Vc* based on a comparison of the genome assemblies to reference sequences of bacterial species in the National Center for Biotechnology Information (NCBI) RefSeq database[49] and the presence of the *Vc*-specific *ompW* outer membrane protein-encoding gene[50]. Among the genomes containing the *ompW* and showing similarity to reference *Vc* sequences, 45 *Vc* genomes had no detectable contamination, i.e., genetic similarity to other pathogens beyond shared accessory genome content, including plasmid sequences, while the other four were mixed with other species. In a total of 19 of 68 cases (~28%), the recovered genomes were associated with non-*Vc* species. In almost 6% (4 of 68 cases), the non-*Vc* isolates were associated with *Aeromonas caviae*, whereas the remainder of the genomes were associated with *Enterobacter cloacae* (~4.4%, 3 of 68 cases), *Providencia alcalifaciens* (~1.5%, 1 of 68 cases), and *Escherichia coli* (~1.5%, 1 of 68 cases), and a mixture of these and/or other bacterial pathogens (~21%, 14 of 68 cases). These observations are consistent with reports elsewhere that other bacterial diarrhoea-associated gastrointestinal pathogens, especially those mimicking *Vc* enteropathy or those poorly investigated via routine laboratory diagnosis, may lead to misdiagnosis of *Vc* in patients with suspected cholera infection[44–48,51–54]. While these findings should be interpreted with caution due to potential

**Table 1 | Summary of cholera cases, deaths, incidences, and case fatality ratio (CFR) per district across Malawi**

| District | Number of cases | Number of deaths | Population size | Overall incidence per 100,000 people | | CFRᵃ |
|---|---|---|---|---|---|---|
| | | | | Cases | Deaths | |
| Mzimbaᵇ | 486 | 3 | 940,184 | 51.69 | 0.3191 | 0.6173 |
| Ntchisi | 68 | 2 | 317,069 | 21.45 | 0.6308 | 2.941 |
| Kasungu | 90 | 6 | 842,953 | 10.68 | 0.7118 | 6.667 |
| Chitipa | 91 | 3 | 234,927 | 38.74 | 1.277 | 3.297 |
| Chikwawa | 748 | 11 | 564,684 | 132.5 | 1.948 | 1.471 |
| Mchinji | 394 | 16 | 602,305 | 65.42 | 2.656 | 4.061 |
| Phalombe | 302 | 14 | 429,450 | 70.32 | 3.26 | 4.636 |
| Zomba | 759 | 28 | 851,737 | 89.11 | 3.287 | 3.689 |
| Mwanza | 120 | 5 | 130,949 | 91.64 | 3.818 | 4.167 |
| Thyolo | 1,520 | 36 | 721,456 | 210.7 | 4.99 | 2.368 |
| Dowa | 1,349 | 39 | 772,569 | 174.6 | 5.048 | 2.891 |
| Mulanje | 936 | 42 | 684,107 | 136.8 | 6.139 | 4.487 |
| Karonga | 1,403 | 25 | 365,028 | 384.4 | 6.849 | 1.782 |
| Nsanje | 607 | 21 | 299,168 | 202.9 | 7.019 | 3.46 |
| Rumphi | 932 | 18 | 229,161 | 406.7 | 7.855 | 1.931 |
| Ntcheu | 1,926 | 56 | 659,608 | 292 | 8.49 | 2.908 |
| Neno | 733 | 14 | 138,291 | 530 | 10.12 | 1.91 |
| Mangochi | 8,182 | 120 | 1,148,611 | 712.3 | 10.45 | 1.467 |
| Dedza | 2,124 | 95 | 830,512 | 255.7 | 11.44 | 4.473 |
| Chiradzulu | 1,426 | 41 | 356,875 | 399.6 | 11.49 | 2.875 |
| Likoma | 204 | 2 | 14,527 | 1404 | 13.77 | 0.9804 |
| Machinga | 2,708 | 104 | 735,438 | 368.2 | 14.14 | 3.84 |
| Nkhotakota | 1,461 | 56 | 393,077 | 371.7 | 14.25 | 3.833 |
| Blantyre | 8,975 | 190 | 1,251,484 | 717.1 | 15.18 | 2.117 |
| Nkhata Bay | 1,628 | 49 | 284,681 | 571.9 | 17.21 | 3.01 |
| Balaka | 4,111 | 81 | 438,379 | 937.8 | 18.48 | 1.97 |
| Salima | 3,591 | 97 | 478,346 | 750.7 | 20.28 | 2.701 |
| Lilongwe | 12,683 | 558 | 2,626,901 | 482.8 | 21.24 | 4.4 |

ᵃCFR Case fatality ratio.
ᵇMzimba shows combined cases from Mzimba South and Mzimba North districts.
The population sizes per district is based on data reported by the National Statistics Office of Malawi for the 2018 population census in Malawi (http://www.nsomalawi.mw/).

misidentification of non-*Vc* enteropathogens, which may be common in settings such as ours, they however suggest the need for further studies to assess the contribution of other non-*Vc* bacteria in cholera-like diarrhoeal diseases during seasonal cholera outbreaks.

### The 2022–2023 outbreak in Malawi was primarily driven by the 7PET lineage O1 Ogawa serotype

We first assessed the distribution of the *Vc* biotype marker genes, widely used to distinguish the classical from El Tor *Vc* lineages, in the 2022–2023 *Vc* isolates from Malawi (Table 2 and Supplementary Data 1 and 2). We found the majority of the 2022–2023 cholera outbreak isolates (~ 96%, 45/47) from Malawi harboured the *ctxB7* and the *rstR* gene alleles typically found in the El Tor *Vc* strains. All the isolates contained the VC2346 gene, a known genetic marker of the 7th pandemic *Vc* strains (Supplementary Data 2). Next, we mapped genomic sequence *k*-mers of the 2022–2023 outbreak and historical *Vc* isolates from Malawi against all known reference LPS O-antigen biosynthesis gene cluster sequences to genotypically determine the specific serogroups, biotypes, and serotypes of the isolates[55]. We found that ~ 95% of 2022–2023 (42 of 44 clinical cases) *Vc* isolates from Malawi sequenced in this study belonged to the O1 serogroup, El Tor biotype, confirming its primary role in the cholera outbreaks in Malawi, consistent with reports from other countries in Africa[7,20,56–59] and elsewhere[60–64] (Fig. 2a). We initially inferred the Ogawa serotype in ~ 80% (35 of 44 cases) and the Inaba serotype in ~ 20% (9 of 44 cases) of the 2022–2023 clinical *Vc* O1 isolates sequenced in this study[65–68]. However, a comparison of the *wbeT* gene of the *Vc* isolates from Malawi inferred to express the Inaba and Ogawa serotypes revealed no specific mutations distinguishing these serotypes, consistent with findings elsewhere[69]. Therefore, we ruled out the presence of Inaba O1 serotype and potential occurrence of capsule-switching events in the 7PET *Vc* isolates as seen in Haiti and Bangladesh[62,69]. In contrast, the two non-O1 *Vc* isolates were assigned serogroups O7 and O19. The low detection rates of non-O1 *Vc* serogroups among the 2022–2023 isolates are consistent with reports from elsewhere that show that cholera outbreaks and epidemics are typically caused by serotypes O1 or O139[12,70]. All the serogroup O1 isolates sequenced in this study (43 of 45 cases), including the environmental isolate, belonged to the 7PET lineage (ST69) widely responsible for seasonal cholera outbreaks and epidemics globally[20,21,71–73]. In contrast, the two non-O1 isolates, O7 and O19, belonged to ST40 and ST635 lineages, respectively (Fig. 2a). Therefore, we concluded that the 7PET O1 serotype Ogawa strains

**Table 2 | Summary of the major attributes of the 2022–2023 and historical *V. cholerae* isolates in Malawi**

| Attribute | 2022–2023 outbreak isolates | Historical isolates (80 s and 90 s) |
|---|---|---|
| Species | *V. cholerae* | *V. cholerae* |
| Serogroup, serotype, and biotype | Mostly O1 Ogawa and a minority of O1 Inaba El Tor and sporadic non-O1/O139 | All O1 Ogawa El Tor |
| Pandemic strain | Pandemic | Pandemic |
| ST or clone | Mostly ST69 (7PET; outbreak); also, ST40 and ST635 (sporadic) | All ST69 (7PET) |
| Genomic wave | Late wave 3 | Early wave 3 |
| Genetic markers | *ctxB7* | *ctxB3* |
| AMR profile | DOXˢ/TETˢ, CIPᴵ, SXTᴿ/SXZᴿ, STRᴿ, CROᴿ, AMPᴿ, CAZᴿ, CARᴿ, CEPᴿ, CPMᴿ, MERˢ, NAᴿ, NITᴿ, STRᴿ, TMPᴿ, FZDᴿ, and CHLᴿ | DOXˢ/TETˢ, CIPˢ, SXTˢ/SXZˢ, STRᴿ, CROᴿ, AMPᴿ, CAZᴿ, CARᴿ, CEPᴿ, CPMᴿ, MERˢ, NAᴿ, NITᴿ, STRᴿ, TMPˢ, FZDˢ, and CHLˢ |
| Horizontally acquired AMR-conferring ICEs | ICEᴳᴱᴺ/ICEVchHai1/ICEVchind5 SXT-like ICEs | None |
| Bacteriophages | CTXφ | CTXφ |
| Pathogenicity islands | VPI-1, VPI-2, and VSP–IIᵃ | VPI-1, VPI-2, and VSP–IIᵇ |

*ST* Sequence type, *AMR* Antimicrobial resistancel, *ICE* Integrative conjugative elements.
ᵃVSP-II pathogenicity island according to GenBank accession no. KM660639.
ᵇVSP-II pathogenicity island according to GenBank accession no. KU601747.

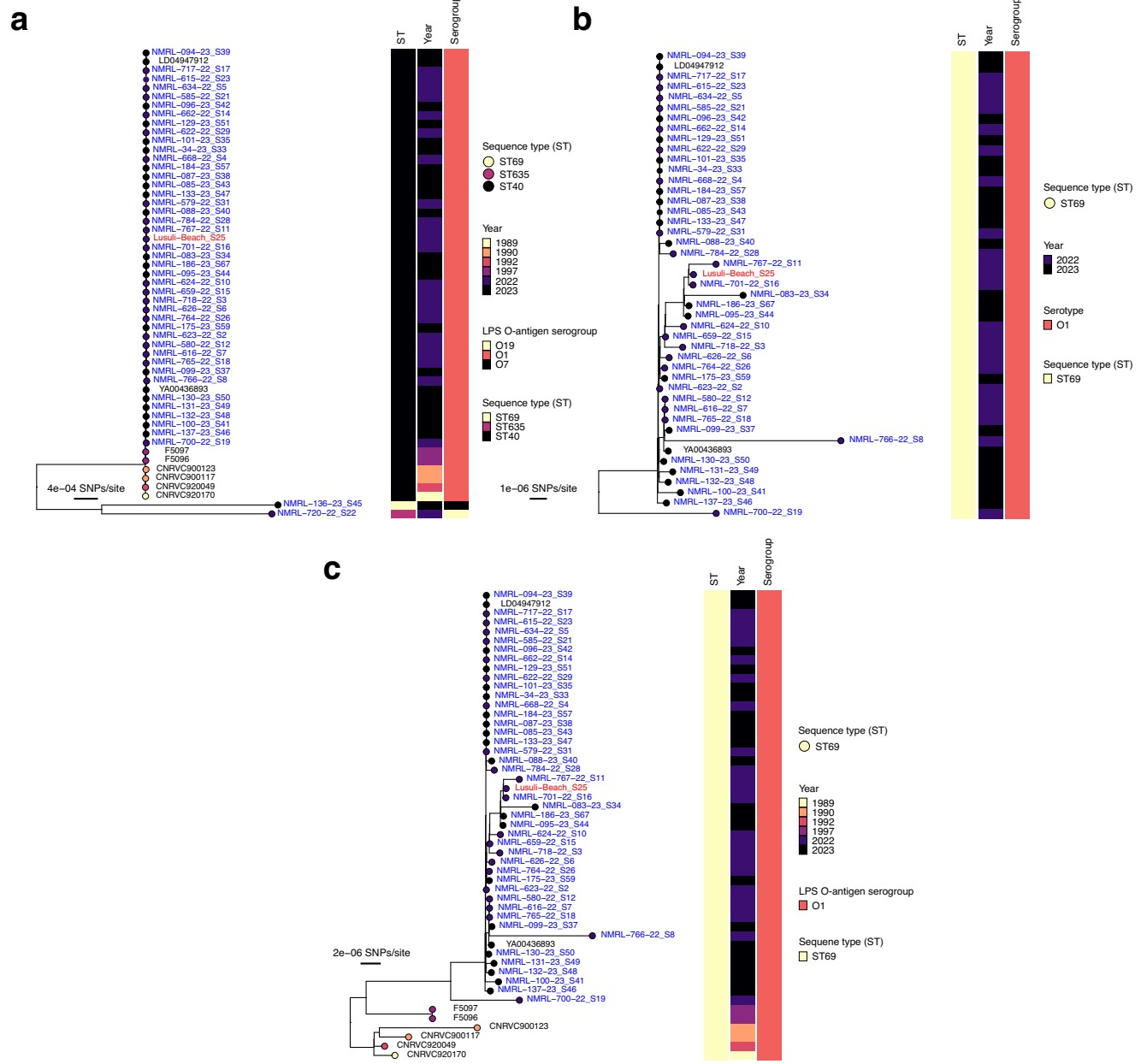

**Fig. 2 | Genetic relatedness of the *V. cholerae* isolates from the 2022–2023 outbreak and the historical isolates from the late 1980s and 1990s in Malawi. a** Maximum likelihood phylogenetic tree showing the genetic relatedness of all the *Vc* isolates from Malawi collected between 2022–2023 in the context of the historical *Vc* isolates from Malawi collected in the late 1980s and 1990s. Two genomes (LD04947912 and YA00436893) represent imported *Vc* into South Africa isolated from people infected in Malawi and are designated as Malawian sequences. **b** Maximum likelihood phylogenetic tree showing the genetic relatedness of the 7PET *Vc* isolates collected between 2022–2023 from Malawi. **c** Maximum likelihood phylogenetic tree showing the genetic relatedness of the 7PET *Vc* isolates collected between 2022–2023 in the context of the historical 7PET *Vc* isolates from Malawi.

The circles with different colours at the tip of the phylogeny represent the year of isolation. All the Malawi 2022–2023 isolates were sampled from human clinical cases (coloured in blue) except one (coloured in red text), which was obtained from a water sample at a beach in the southern region of Malawi. The phylogeny is annotated by colour strips at the tips of each tree representing the sequence type (ST), year of isolation, and LPS O-antigen serogroup. The phylogeny was constructed based on the core-genome SNPs identified from the merged alignments of chromosomes 1 and 2, and rooted based on an outgroup *Vibrio mimicus* species, not shown in the tree. The isolates with taxon labels coloured in black in all the phylogenetic trees were sequenced and reported by previous studies (Supplementary Data 1).

predominantly drove the 2022–2023 cholera outbreak in Malawi, and the non-O1 isolates were likely associated with sporadic seasonal cases.

### Genetic analysis suggests the 2022–2023 outbreak-associated *V. cholera* isolates were recently imported from Asia

To investigate the potential origin of the Malawi 2022–2023 outbreak-associated *Vc* isolates, we placed the isolates in the global phylogeny in the context of genomes from Malawi and other countries. We obtained

a large collection of 2379 globally diverse *Vc* sequences from 110 countries or territories worldwide (see methods). Our collection included *Vc* sequences from notable recent and past cholera outbreaks and epidemics globally, including other African settings, which captured the global *Vc* genetic diversity (Table 2 and Supplementary Data 1). We compared the non-7PET Malawian isolates deemed to cause sporadic cases during the 2022–2023 outbreak, belonging to ST40 and ST635 clones. The ST40 and ST635 isolates showed closest

genetic similarity with a single isolate from India and three isolates from Austria of identical STs, respectively. The single ST40 isolate from India was collected in 1962 and differed from the 2023 Malawi isolate by ~596 SNPs, while ~5299 SNPs separated the 2022 Malawi ST635 isolate from the three recent Austrian ST635 isolates sampled in 2012.

In contrast, a maximum likelihood phylogeny of the Malawi *Vc* isolates in the context of the global 7PET *Vc* isolates based on 14,557 SNPs (20,993 SNPs before excluding recombinogenic regions) revealed that the predominant 2022–2023 outbreak-associated 7PET isolates from Malawi formed a single cluster in the phylogeny (Fig. 3). An interactive phylogeny of the 7PET sequences is available on Microreact (https://microreact.org/project/malawi-vibrio-cholerae-2022-2023-outbreak). These 2022–2023 Malawi isolates showed the closest genetic similarity (~11–16 SNPs) to isolates from Pakistan (2022) and Iraq (2022)[74] which represents a recent transmission from Asia into Africa recently designated T15 or AFR15 lineage[23]. The most recent common ancestor of the 2022–2023 Malawi outbreak isolates and those associated with cholera activity in Pakistan and Iraq in 2022 appeared to be of Asian origin, from India and Bangladesh[7,8]. However, based on the global 7PET phylogeny, one slightly divergent Malawian isolate from the 2022–2023 outbreak (sample name: NMRL-700-22_S19) shared the most recent common ancestor with T11 *Vc* isolates collected in 2015 in Zimbabwe. Further genomic comparison of the Malawi 2022–2023 *Vc* outbreak isolates to the historical O1 serotype

Inaba 7PET strains collected from the late 1980s and 1990s revealed up to ~228–479 SNP differences and the historical isolates were associated with earlier transmission sublineages (T1 and T5)[7] from Asia and were not the most recent ancestor of the current outbreak strains (Fig. 3). The historical T1 7PET isolates from Malawi clustered with sequences from Angola and Mozambique T5 7PET isolates clustered with isolates from Zambia, Tanzania, and South Africa. Together, these findings suggest that the predominant outbreak-associated strain associated with the 2022–2023 cholera outbreak in Malawi is a highly successful clone, which has been disseminated globally in the past two decades following its emergence in Asia[7,8], and appears to be a new recent importation of *Vc* strains from Asia rather than a re-emergence of a previously circulating strain in Africa.

## The 2022–2023 outbreak-associated *V. cholerae* isolates from Malawi harbour a diverse set of virulence genes

One of the strongest prevailing hypotheses for the occurrence of the largest cholera outbreak in Malawi from 2022–2023 primarily pertains to tropical cyclones Ana and Gombe that occurred in early 2022. However, although similar cyclones occurred in some years prior to 2022, the reported cholera cases and deaths during each corresponding seasonal outbreak were significantly lower than seen during the 2022–2023 outbreak[37]. Understanding the environmental and pathogen-specific factors which may

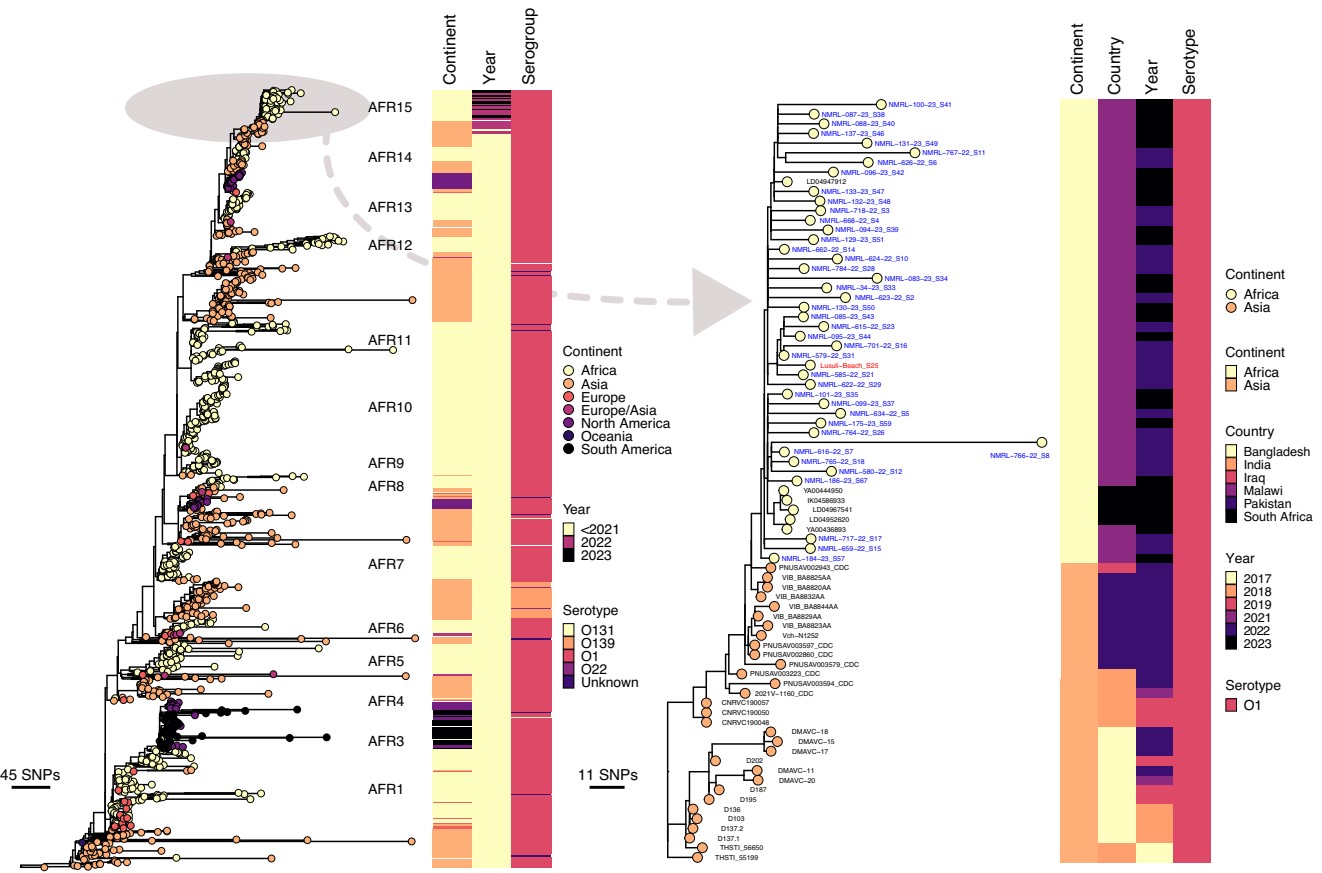

**Fig. 3 | Maximum likelihood phylogenetic tree showing genetic relatedness of the 2022–2023 and historical Malawi 7PET *Vc* isolates in the context of global 7PET *Vc* genomes.** The phylogeny was constructed based on the core-genome SNPs identified after removing recombinogenic regions from the merged alignments of chromosomes 1 and 2, and rooted based on the A6 genome (GenBank assembly accession: GCF_001255575.1) as an outgroup. An interactive phylogenetic tree of the current outbreak and historical 7PET *Vc* isolates from Malawi in the context of the global 7PET sequences is available on Microreact (https://microreact.

org/project/malawi-vibrio-cholerae-2022-2023-outbreak). All the Malawi 2022–2023 isolates were sampled from human clinical cases (coloured in blue) except one (coloured in red text), which was obtained from a water sample at a beach in the southern region of Malawi. The isolates with taxon labels coloured in black in all the phylogenetic trees were sequenced and reported by previous studies (Supplementary Data 2). Two genomes (LD04947912 and YA00436893) represent imported *Vc* into South Africa isolated from people infected in Malawi and are designated as Malawian sequences.

contribute to the high incidence of cholera cases and deaths during the 2022–2023 cholera outbreak in Malawi is critical to minimise the impact of future outbreaks. We hypothesised that the presence of a diverse set of virulence genes in the 2022–2023 outbreak-associated *Vc* isolates, as similarly seen in other isolates linked with large outbreaks and epidemics elsewhere, including Haiti in 2010[75] and Yemen in 2016–2017[76], might promote their pathogenicity and virulence. Our analysis of the virulence and pathogenicity profiles of the 2022–2023 Malawi cholera outbreak *Vc* isolates to assess the presence and absence patterns of genes and MGEs known to be associated with the pathogenicity and virulence of *Vc* strains revealed similar distribution of these genes with the 7PET isolates elsewhere (Table 2 and Supplementary Data 2). These findings suggest that the 2022–2023 *Vc* isolates from Malawi did not harbour a unique set of genetic factors that may have resulted in enhanced virulence compared to the virulence factors commonly found in other 7PET *Vc* strains typically associated with global outbreaks.

### The 2022–2023 outbreak-associated *V. cholerae* isolates show higher AMR and harbour SXT/R391-like ICE than the historical isolates from Malawi

We screened for the presence of genotypic antibiotic resistance for seventeen antibiotics (Fig. 4b). We found higher genotypic resistance in 7PET than in non-7PET isolates, associated with nine to fifteen antibiotics. The 7PET isolates showed tetracycline and doxycycline (TET/DOX) susceptibility; this antibiotic is a recommended treatment option for severe cholera infection that complements the polymer-based or glucose-based oral rehydration therapy[4,5]. We also observed intermediate resistance against ciprofloxacin (CIP), which is an alternative treatment option in children in Malawi, conferred by point mutations in the *gyrA* and *parC* genes[77]. Interestingly, the historical 7PET isolates from Malawi showed resistance to fewer antibiotics than the 2022–2023 outbreak isolates, which consistently showed genotypic resistance to the antibiotics mentioned above, except for CIP, TET, and meropenem (MER) (Fig. 4b, Supplementary Data 2).

Nearly all the Malawi 2022–2023 outbreak-associated isolates sequenced in this study harboured *aph(3″)*-Ib and *aph(6)*-Id, and the dual presence of *strA* and *strB*, which contributed to the streptomycin (STR) resistance (Supplementary Data 2). We noted universal trimethoprim (TPM) resistance conferred by the *dfrA1* gene in the current outbreak-associated 7PET *Vc* isolates, but non-7PET isolates from the current outbreak did not harbour this gene. We also detected sulfonamide (sulfamethoxazole [SXT] and sulfisoxazole [SXZ]) resistance gene *sul2* and the SXT-like ICE-borne chloramphenicol (CHL) resistance gene *floR* in the Malawi 2022–2023 outbreak-associated *Vc* 7PET isolates. Additionally, we found the *catB9* gene in the current outbreak 7PET isolates although it does not confer CHL resistance in *Vc*[78]. However, we did not find any tetracycline resistance genes carried by the SXT-like ICEs in all the isolates from Malawi, which suggested universal susceptibility of the *Vc* isolates to TET/DOX, consistent with the findings from other African countries, including from Kenya[79], Central African Republic[56], and Algeria[57]. Although we observed universal carbapenems (CAR) genotypic resistance in the current outbreak isolates from Malawi, further phenotypic characterisation of the 7PET isolates from Malawi is needed to determine the underpinning resistance mechanisms[80].

We then performed an in-depth analysis of the 2022–2023 outbreak and historical *Vc* isolates from Malawi to identify the SXT/R391-like ICEs characteristic of global 7PET wave 3 *Vc* isolates[8] (Fig. 4b, Supplementary Data 2). Our *k*-mer-based mapping analysis showed the absence of all the versions of the SXT/R391-like ICEs in the historical and non-7PET *Vc* isolates collected during the 2022–2023 outbreak in Malawi consistent with the distribution of the ICE-borne AMR genes. In contrast, all the outbreak-associated 7PET isolates contained an SXT/

R391-like element genetically closest to the ICE[GEN], ICEVchHai1, ICEVchind5, and ICEVchban5 ICEs (Fig. 4b, Supplementary Data 2). All the outbreak-associated 7PET *Vc* isolates showed the highest mapping coverage against the ICEVchban5, ICEVchHai1, and ICE[GEN] reference sequences (Supplementary Data 2). These ICEs are commonly found in outbreak-associated *Vc* isolates of Asian origin, including those reported in Africa, Bangladesh, India, Nepal, Yemen, and Haiti[7,8,21,60,62].

Together, these findings suggest that the *Vc* strains responsible for the 2022–2023 cholera outbreak in Malawi are genotypically resistant to more antibiotics than the historical *Vc* isolates from Malawi due to the presence of the ICE[GEN]/ICEVchHai1/ICEVchind5 SXT/R391-like ICEs. However, the absence of TET/DOX resistance genes supports the continued use of tetracyclines in Malawi as a first-line antibiotic for treating cholera-affected patients.

## Discussion

Cholera outbreaks and epidemics associated with the seventh pandemic, exemplified by those in 2010–2011 Haiti[81] and 2016–2017 Yemen[9], continue to cause a significant diarrhoea-associated disease burden globally, especially in endemic settings in the low-and-middle-income countries (LMIC)[43,82,83]. Here, we describe that the deadliest, and 2022–2023 cholera outbreak in Malawi's history, which started in March 2022, is caused by the O1 serogroup El Tor biovar. We identified co-circulation of the Ogawa and Inaba serotypes, highlighting potential serotype switching as seen elsewhere[62,69], with a predominance of the Ogawa serotype, which appears to be the dominant serotype in the African region[7,20,56–59] and other continents[60–64]. In contrast to the previous seasonal cholera outbreaks in Malawi[37,43], the 2022–2023 outbreak caused cholera cases in all districts, with the persistence of cases even during the dry season throughout 2022 in some districts, including Blantyre, Chikwawa, and Neno. We speculate that the persistence of cases in these districts, particularly Blantyre, which is a major city and transportation hub linking the southern region to other districts in Malawi, may have contributed to the human-to-human *Vc* transmission to other districts following the onset of the outbreak in Nsanje district.

Despite identifying diverse *Vc* lineages, associated with three STs, the 2022–2023 outbreak was primarily driven by the 7PET lineage, which showed the highest similarity to the *Vc* strains associated with cholera activity in Iraq[23] and Pakistan in 2022[74], suggesting a potential Asian origin, consistent with recent phylogeography work[23,84]. These findings further support the suggestions of a new recent importation of a *Vc* sublineage AFR15 or T15 into Malawi and Africa from Asia rather than a re-emergence of a previously circulating strain in Africa[23]. Altogether, these findings demonstrate that the 2022–2023 cholera outbreak in Malawi is caused by recently imported multi-drug resistant 7PET O1 *Vc* strains susceptible to tetracycline, highlighting the importance of genomic surveillance in elucidating the genetic makeup of *Vc* strains to understand the evolution and spread of the *Vc* strains, especially in monitoring the emergence and spread of antimicrobial resistance to first-line treatment options. While there is a possibility that the Malawi 2022–2023 outbreak strain may have circulated locally and in the region, this is difficult to prove decisively due to the limited availability of genomic data from Africa. Continued genomic surveillance efforts are critical for understanding the emergence and long-term *Vc* strain dynamics in the context of oral cholera vaccine rollout in Malawi[85] and other settings[86,87] strategies for controlling and preventing cholera outbreaks.

Previous studies in Africa and elsewhere have suggested that some enteric bacteria may cause cholera-like diarrhoea infections, which may be incorrectly attributed to *Vc* during cholera outbreaks[44–48]. Although our study was not designed to test the proportion of cholera-like cases attributed to non-*Vc* enteric pathogens, our genomic analysis identified these bacterial pathogens in ~28% of the suspected *Vc* isolates collected from patients presenting with

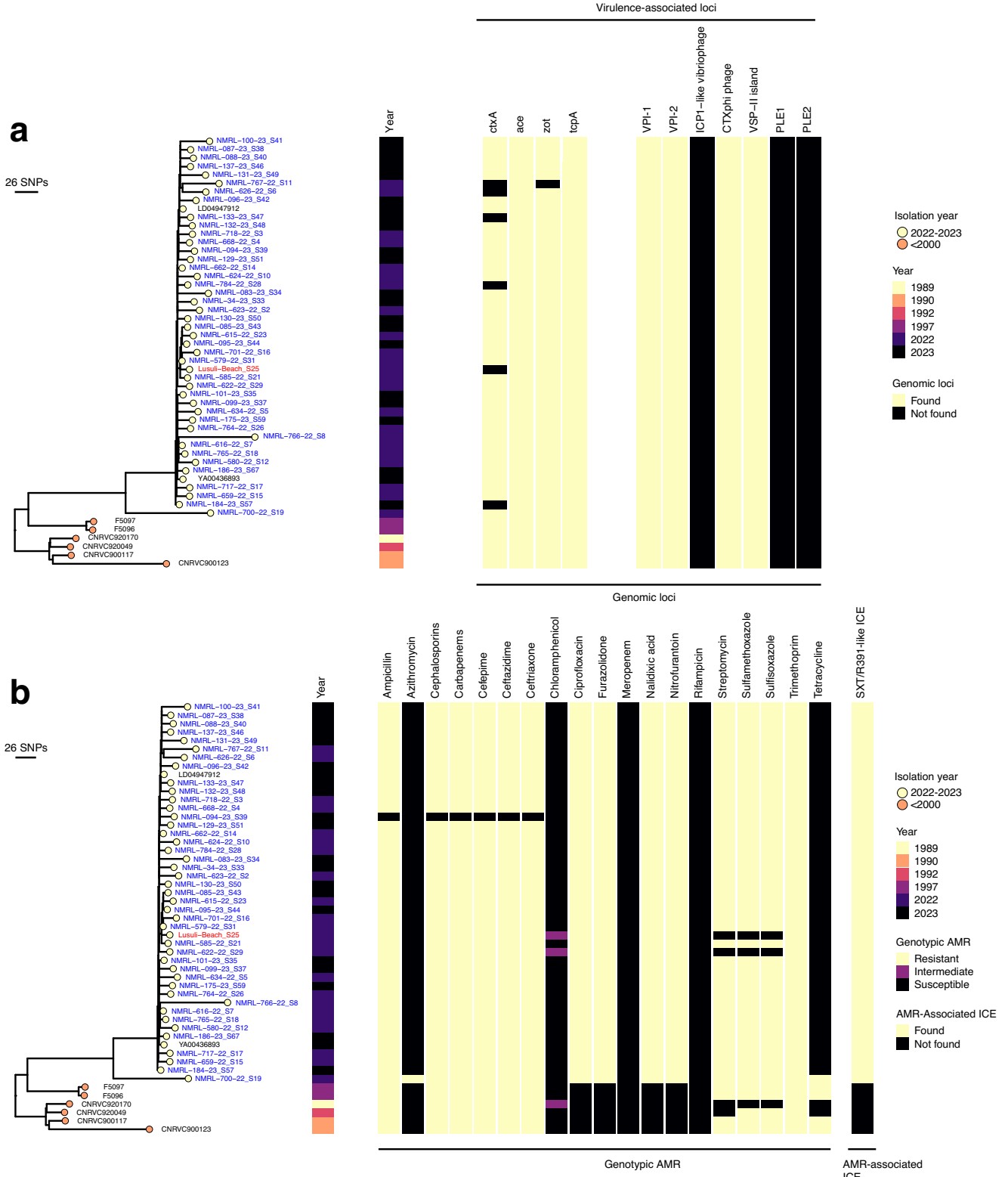

cholera-like symptoms. These findings suggest that it is possible that profuse diarrhea cases can be incorrectly attributed to *Vc*. This is especially true in sub-Saharan African settings, such as Malawi, where a full battery of readily available tests to accurately identify a range of non-*Vc* enteropathogens may not be routinely available. These findings have repercussions towards patient management and could explain, in part, why the 2022–2023 outbreak persisted. Although we recovered several enteric bacteria in *Vc*-negative samples, *Aeromonas caviae*, which mimics *Vc* enteropathy[44–48,51–54], maybe the most common non-*Vc* bacterial pathogen in these incorrectly diagnosed cholera-like diarrhoea cases. Our literature search revealed no epidemiological studies on the fraction of cholera-like cases attributable to *Aeromonas spp*. in sub-Saharan Africa. Thus, to inform public health responses, further studies are required to understand the proportion of cholera-like cases attributable to *Vc* and other enteropathogens during cholera outbreaks.

**Fig. 4 | Phylogenetic distribution of the virulence factors and genotypic anti-microbial resistance in the outbreak-associated 7PET isolates from Malawi.** **a** Distribution of virulence genes, pathogenicity islands, and bacteriophages in the 2022–2023 and historical *Vc* isolates from Malawi. The full list of virulence genes present in the Malawi cholera outbreak isolates is available in Supplementary Data 2. **b** Distribution of genotypic AMR profiles and presence and absence of the SXT/R391-like ICE in the 2022–2023 and historical *Vc* isolates from Malawi. The circles with different colours at the tip of the phylogeny represent the year of isolation. All the Malawi 2022–2023 isolates were sampled from human clinical cases (coloured in blue) except one (coloured in red text), which was obtained from a water sample at a beach in the southern region of Malawi. The phylogeny is annotated by colour strips at the tips of each tree representing the location of origin (continent), year of isolation, and LPS O-antigen serogroup. The phylogeny was constructed based on the core-genome SNPs identified after removing recombinogenic regions from the merged alignments of chromosomes 1 and 2, and rooted based on the A6 genome (GenBank assembly accession: GCF_001255575.1) as an outgroup. Two genomes (LD04947912 and YA00436893) represent imported *Vc* into South Africa isolated from people infected in Malawi, as such are designated as Malawian sequences. The isolates with taxon labels coloured in black in all the phylogenetic trees were sequenced and reported by previous studies (Supplementary Data 2). The presence of the CTXφ prophage was confirmed by mapping the reads of each *Vc* isolate against the sequence of an intact CTXφ prophage (see methods).

One of the plausible hypotheses that could explain the increased risk of cholera transmission are humanitarian crises that disrupt water and sanitation systems, displace populations towards inadequate and overcrowded camps, and climatic conditions such as flooding due to cyclones. Malawi has experienced three tropical cyclones since January 2022 (Ana in January 2022, Gombe in March 2022, and Freddy in March 2023). The first cholera wave was reported in the Nsanje district in March 2022, which coincided with the occurrence of tropical cyclone Gombe in Malawi, which caused floods in the lower Shire area in the southern region of Malawi. It is likely that the disruption to water supplies and sanitation facilities, as well as overcrowding in camps caused by tropical cyclone Ana, which occurred in the southern region of Malawi approximately two months before cyclone Gombe, created a conducive environment to kickstart the cholera outbreak. However, human-to-human transmission events likely contributed to the spread of cholera across the country throughout the dry season in 2022, which is strikingly different from the past years, where cholera outbreaks have occurred only during the rainy seasons. Notably, cholera cases in all districts of the northern region peaked during the hot months in 2022, contrary to the central region and most districts in the southern region (except Nsanje, Neno, Chikwawa, and Blantyre), where most of the cases were reported after November 2022 subsequently peaking in early 2023. Surprisingly, the occurrence of cyclone Freddy in early 2023 coincided with a reduction in cholera cases in all districts except the two lower shire river districts, Chikwawa and Nsanje, which are typically affected by floods and tend to kickstart cholera outbreaks in Malawi. Our findings suggest that the occurrence of the tropical cyclones in Malawi may have precipitated the deadliest cholera outbreak in Malawi, highlighting the impact of climatic changes on the risk of cholera and possibly other infectious diseases.

Despite the occurrence of previous cyclones in Malawi, the number of cholera cases has never reached the magnitude seen during the 2022–2023 outbreak. Before 2022, the two largest cholera outbreaks in Malawi occurred between 1998–1999 (CFR = 3.4%; 860 of 25,000 cases) and 2001–2002 (CFR = 2.3%; 968 of 33,546)[37]. Therefore, we hypothesised that in addition to the climatic conditions conducive to *Vc* transmission and infection, additional *Vc*-specific factors might explain the scale of the 2022–2023 cholera outbreak in Malawi. Our comparative genomic analysis of the 2022–2023 and historical *Vc* isolates from Malawi in the context of global *Vc* sequences revealed two potential independent introductions of the 2022–2023 strains into Malawi. The close genetic similarity of the Malawi isolates to those from Asian countries, including Bangladesh, India, Yemen, and also Haiti, which were originally imported from Asia[60], suggested that the 2022–2023 outbreak 7PET clone, serogroup O1 strains may have also originated from Asia[7,8]. In terms of the virulence profiles, the 2022–2023 outbreak-associated O1 *Vc* strains in Malawi mostly exhibit similar characteristics to the historical Malawian *Vc* isolates. The major distinguishing genetic characteristic of the 2022–2023 isolates from the historical isolates is the presence of a different version of the VSP-II pathogenicity island and *ctxB7* genotype, although it's unlikely that these differences may partly explain the observed transmission and virulence of the 2022–2023 O1 strains in Malawi. Collectively, these findings emphasise the critical role of natural disasters, such as earthquakes and cyclones, and humanitarian crises, such as wars, which displace individuals and disrupt water supplies and sanitation systems, in potentiating cholera outbreaks, even more so than the bacterial factors themselves. This highlights the need for investing in quality water supply and sanitation systems and other measures to minimise the occurrence of humanitarian crises, preparedness for environmental disasters, and the availability of oral cholera vaccines to reduce the risk of cholera outbreaks.

We acknowledge some limitations. First, the burden of cholera during the 2022–2023 outbreak might have been underestimated due to underreporting of cases because of the unavailability of diagnostic tests, especially in rural settings, and negative healthcare-seeking behaviour, mostly among those presenting with mild disease. Second, we did not employ a full battery of assays required to accurately identify non-*Vc* enteropathogens, which may cause cholera-like symptoms, and to phenotypically determine specific O1 serotypes. However, the identification of non-*Vc* bacterial isolates associated with cholera-like diarrhoea has been reported even in high-income settings, suggesting that the isolation of these bacteria, especially *Aeromonas spp.*, may carry clinical relevance. Therefore, we recommend further studies to investigate the contribution of non-*Vc* bacteria in cholera-like diarrhoea during seasonal cholera outbreaks in sub-Saharan Africa. Third, there are limited genotypic characteristics of *Vc* strains in Malawi, other African, and other LMIC countries beyond the country-specific case counts tracked by the WHO. In this study, we had access to only a few contextual *Vc* genomes from Malawi collected in the late 1980s and 1990s, which limited our ability to determine temporal changes in the distribution of *Vc* strains leading to the 2022–2023 outbreak. Consequently, whole genome sequences generated in our study will start to close this knowledge gap and will be critical in providing the much-needed context to understand the origin of *Vc* strains associated with future outbreaks in Malawi. Fourth, due to the small number of sequenced genomes, we could not compare the characteristics of *Vc* strains collected from different districts or regions and temporal scales in Malawi and assess their association with the clinical characteristics, including disease incidence and CFR. Improved disease surveillance systems, particularly sample collection, preservation, and tracking are critical to generating robust genomic and epidemiological insights in Malawi. Furthermore, we did not perform extensive sampling of environmental *Vc* isolates as well as phylogeography and phylodynamic analyses in this study. Follow-up work should include *Vc* from the environmental sources and conduct these more elaborate phylogenomic analyses using BEAST[88] and other tools[89–91], to estimate the frequency and timing of *Vc* O1 7PET lineage into Malawi, as recently reported elsewhere[84]. The use of long-read sequencing in future studies would be useful to gain better insights into the structural variation in the *Vc* pathogenicity islands, ICEs, and prophage sequences, beyond variation at the SNP level.

Our study has provided an early snapshot of the genomic characteristics associated with the 2022–2023 *Vc* outbreak in Malawi, the

deadliest cholera outbreak ever recorded in the country. Our whole-genome sequencing of *Vc* isolates collected across Malawi shows that the 2022–2023 cholera outbreak is driven by a newly imported late wave 3 serotype Ogawa 7PET strains belonging to the AFR15 or T15 introduction event, rather than strains derived through stepwise evolution from the historical local serogroup O1 *Vc* strains. The combination of the devastating cyclones and the introduction of a new *Vc* strain in Malawi offered a perfect opportunity for the outbreak. This work highlights a concerted locally-driven genomic surveillance effort, with support from international partners, to understand the genomic epidemiology of *Vc* strains linked with the 2022–2023 outbreak. Continued molecular and genomic surveillance in Malawi and the region will be crucial to understanding long-term strain dynamics, including the impact of oral cholera vaccines, AMR, and the geographical spread of *Vc*.

## Methods

### Ethical approval
This work was conducted according to the guidelines of the Declaration of Helsinki and was approved by the National Health Sciences Research Committee, Lilongwe, Malawi (Protocol #867) and the Research Ethics Committee of the University of Liverpool, Liverpool, UK (000490) under the Diarrhoea Surveillance study, and the College of Medicine Ethics Committee (COMREC, Protocol #P.10/22/3790) under the NIHR Global Health Research Group on Gastrointestinal Infections: Facilitating the Introduction and Evaluation of Vaccines for Enteric Diseases in Children in Eastern and Southern sub-Saharan Africa study. For the present study patient data were anonymized. Additional consent is not required for studying submitted strains with anonymised patient data.

### Analysis of Malawian cholera cases and deaths
We analysed the case and deaths data obtained from the Public Health Institute of Malawi (PHIM), Malawi Ministry of Health data on May 20, 2023 (https://cholera.health.gov.mw/surveillance) using the dplyr (version 1.0.8) package for data wrangling (https://github.com/tidyverse/dplyr) in R (version 4.0.3) (https://www.R-project.org/). We calculated the incidence of cases and deaths by dividing the total number of cases reported in Malawi or per district by the population size multiplied by 100,000. We used the population sizes per district based on data reported by the National Statistics Office of Malawi for the 2018 population census (http://www.nsomalawi.mw/). To calculate the CFR during the course of the 2022–2023 cholera outbreak, we first obtained the number of cases and deaths using a sliding window of 21 days using the runner (version 0.3.7) package (https://CRAN.R-project.org/package=runner) and then calculated the percentage of deaths during each window. We plotted the number and incidence of cases and deaths per district using sf (version 1.0.7), rnaturalearth (version 0.3.2), rnaturalearthdata (version 0.1.0), and rnaturalearthhires (version 0.2.1) packages in R (version 4.0.3) (https://www.R-project.org/).

### Specimen collection and preparation
Since surveillance of *Vc*, especially the preservation of clinical isolates is not routinely undertaken in all districts in Malawi, we could not systematically select representative samples for microbiological examination. Faecal specimens (liquid stools) were previously collected in clean unchlorinated disposable containers from patients presenting with cholera-like symptoms, including profuse watery diarrhoea and vomiting, at Cholera Treatment Units (CTUs) in Malawi from March 2022 to February 2023. A stool-soaked rectal swab was placed into a Cary-Blair transport medium (Oxoid, Thermo Fisher Scientific, USA) and transported to the hospital laboratories within two hours of collection. Once the stool samples arrived in the laboratory, a cholera Rapid Diagnostic Test (RDT) and culture were performed. In addition, a stool-soaked swab from each sample was then inoculated in Alkaline Peptone Water (APW) (Becton Dickinson, UK) for enrichment,

which was kept between 4–6 h at ambient temperature prior to RDT and culture. We did not use clinical metadata related to the patients, and all isolate identifiers were de-identified; therefore, additional institutional review board approval was not required.

### A rapid diagnostic test procedure for *V. cholerae*
Testing was performed by qualified and well-trained laboratory technologists from the District Hospital Laboratories and the Malawi National Microbiology Reference Laboratory. In brief, Crystal VC Ag O1/O139 rapid diagnostic testing kits were used for testing samples and interpreting results by following the manufacturer's instructions (Arkray, Japan). About four drops of watery stool were transferred into a sample processing vial (pre-filled with 1 ml of sample diluent buffer), and the Crystal VC strip was dipped into it for at least 15 min; the test line and/or control line appeared as a red colour. The appearance of both lines indicated that the sample was positive for *Vc* serogroup O1; the appearance of only the control line but not the test line indicated a negative result for the test.

### Identification of *V. cholerae*
Strict laboratory safety precautions were followed when working with suspected cholera specimens. Appropriate personal protective equipment (PPEs) was always worn, and standard precautions were followed for handling and disposing of biological materials. The stool samples were streaked (cultured) on Thiosulphate Citrate Bile Sucrose Agar (TCBS) (Becton Dickinson, UK) media after four hours of incubation in APW, and then incubated at 37 °C for 18–24 h. After 18–24 h of incubation on TCBS, large (2–4 mm in diameter) slightly flattened, yellow colonies with opaque centres and translucent peripheries were examined, suggestive of *Vc*. Well-isolated (pure) single yellow colonies were picked and streaked on Nutrient Agar (Becton Dickinson, UK) and incubated at 37 °C for 24 h. Presumptive identification of *V. cholerae* was made with a positive oxidase biochemical test.

### Bacterial DNA extraction
Genomic DNA of the suspected *Vc* colonies was extracted using QIAamp DNA Mini Kit (Qiagen, Germany), at the National Microbiology Reference Laboratory within the Public Health Institute of Malawi (PHIM). The extracted nucleic acid material was shipped on dry ice to the University of the Free State-Next Generation Sequencing (UFS-NGS) Unit, Bloemfontein, South Africa, for library preparations and whole genome sequencing.

### Library preparation
The bacterial DNA samples were quantified on a Qubit fluorometer using a High Sensitivity dsDNA Assay kit (Thermo Fisher Scientific, USA). The obtained DNA concentrations were normalised to 0.2–0.3 ng/µl by diluting with an elution buffer (Qiagen, Germany). Genomic libraries were prepared with the Nextera XT DNA Library preparations kit (Illumina, USA). Normalized DNA was enzymatically fragmented and simultaneously tagged with Illumina sequencing adapters, and each sample of the fragmented DNA was uniquely indexed using Nextera DNA CD Indexes (Illumina, USA). This was followed by library size selection and purification using Ampure XP magnetic beads (Beckman Coulter, USA) and freshly prepared 80% ethanol.

### Library validation and sequencing
The quality of the libraries and fragment size distribution was assessed on Agilent 2100 Bioanalyzer using the dsDNA High Sensitivity Assay kit (Agilent Technologies, USA), and the average fragment size obtained was 600 bp. The purified libraries were fluorometrically quantified on Qubit 3.0 fluorometer, followed by normalisation to equimolar concentrations of 4 nM. Normalized libraries were pooled into a clean 1.5 ml tube. The library pool was denatured with a freshly prepared 0.2 N sodium hydroxide (NaOH), followed by dilution with a

hybridisation buffer (HT1) to a final concentration of 8 pM. Lastly, the library was spiked with 0.5% PhiX sequencing control (20 pM), and DNA sequencing was performed on a MiSeq platform (Illumina) for 600 cycles, using a V3 reagent kit (Illumina, USA), to generate $2 \times 301$ pb paired-end reads. After sequencing, the libraries were demultiplexed on an instrument based on the unique index sequences and separate fastq files that were generated for each library. In summary, nearly 80% of the data achieved a Phred score of at least Q30 and a cluster density of 926 K/mm$^2$ was achieved, with 88.7% of the clusters having passed the filter. We used cutadapt (version v4.4) to trim adapters from the raw sequence reads[92].

## Comparative genomic analysis

The generated sequence reads were assembled using SPAdes (version 3.14.0)[93]. The species assignment was done using Kraken (version 2.1.2)[94]. To generate the whole-genome phylogeny of the *Vc* isolates, we first mapped the assembled contigs of each *Vc* isolate and an outgroup *Vibrio mimicus* genome Y4 strain (GenBank accessions: CP077425 and CP077426) against a merged reference sequence of *Vc* O1 biovar El or strain N16961 chromosome 1 (GenBank accession: AE003852) and 2 (GenBank accession: AE003853) separated by ambiguous bases (Ns) using Snippy (version 4.6.0) (https://github.com/tseemann/snippy). We used the "−ctgs" option to determine SNPs between the assembled contigs and the merged reference genome. We then compared the merged reference sequence against nucleotide sequences of known *Vc* pathogenicity islands, prophages, and ICEs to identify their genomic coordinates using BLASTN (version 2.12.0 + )[95]. We then masked the genomic regions containing pathogenicity islands, prophages, and ICEs using 'maskfasta' option implemented in bedtools (version 2.30.0)[96]. We identified variable sites in the whole-genome alignment containing SNPs using snp-sites (version 2.5.1)[97,98], which resulted in the final core-genome SNP alignment used for phylogenetic analysis.

To place the Malawian *Vc* genomes in the global context, we obtained a large collection of globally diverse *Vc* isolates worldwide available in the VibrioWatch implemented in the PathogenWatch web tool (https://pathogen.watch/) and recent reported genomes from Smith et al.[23] We constructed a maximum likelihood core-genome phylogeny of the Malawi and contextual global *Vc* isolates using IQTREE (version 2.0.3)[99] using the best-fit model selection. We rooted the generated global phylogeny containing both 7PET and non-7PET *Vc* strains on the branch separating the *Vc* isolates and the outgroup *V. mimicus* genome using the "root" function in the APE package (version 5.6.2)[100]. Next, we ladderised the outgroup-rooted phylogeny using "ladderize" function in APE (version 5.6.2)[100], and then dropped the outgroup taxon from the phylogeny, for clarity, using the "drop.tip" function in APE (version 5.6.2)[100]. We exported a Newick file of the resulting phylogenetic tree and visually explored it using Taxonium, a web-based phylogenetic visualisation tool (https://taxonium.org/)[101]. To further understand the evolution of the 7PET strains, we randomly selected isolates selected isolates from each country per year. We generated a whole-genome alignment for these isolates as described in the previous paragraph and then run Gubbins (version 3.2.1)[102] to detect and remove recombination events from the alignment and generate a recombination-free phylogeny using IQTREE (version 2.0.3)[99]. We specified the A6 genome (GenBank assembly accession: GCF_001255575.1) as the outgroup when performing the recombination analysis with Gubbins. To show the genetic relatedness of specific *Vc* isolates, we pruned the global phylogeny and performed additional visualisation using the APE package (version 5.6.2)[100] and ggtree (version 3.10.0)[103]. We determined the number of core-genome SNPs distinguishing specific *Vc* isolates using snp-dists (version 0.8.2) (https://github.com/tseemann/snp-dists).

To assess the presence and absence of genes used for species identification and biotyping, and those encoding virulence factors and AMR, we used ABRicate (version 1.0.1) (https://github.com/tseemann/abricate) with "--minid 75 --mincov 75" options. We compared each genome to a reference database of genes obtained from the NCBI AMRFinderPlus database and the virulence factor database (VFDB)[104]. We used a custom database of the genes used for biotyping, AMR, and pandemic lineage and species identification in the CholeraeFinder tool (https://cge.cbs.dtu.dk/services/CholeraeFinder). We obtained these genes from the tool's repository (https://bitbucket.org/genomicepidemiology/choleraefinder_db/src/master/). The presence of the VC2346 gene (GenBank accession: AE003852) was used to determine pandemic *Vc* lineages while *rstR* classical (GenBank accession: KJ023707), *rstR* El Tor (GenBank accession: AE003852), *tcpA* classical (GenBank accession: M33514), *tcpA* classical (GenBank accession: CP001235), *tcpA* El Tor wave 3 (GenBank accession: AF325734) were used to assign the classical and El Tor biotypes. We identified specific *ctxB* genotypes of the *Vc* isolates by comparing the assemblies to *ctxB1* (GenBank accession: CP001235), *ctxB3* (GenBank accession: AE003852), and *ctxB7* (GenBank accession: JN806157) gene sequences. Additionally, we also used VibrioWatch implemented in the PathogenWatch web tool (https://pathogen.watch/) to infer the genotypic AMR of the *Vc* isolates for fourteen antibiotics, namely, ceftriaxone (CRO), ampicillin (AMP), ceftazidime (CAZ), carbapenems (CAR), cephalosporins (CEP), chloramphenicol (CHL), ciprofloxacin (CIP), cefepime (CPM), meropenem (MER), nalidixic acid (NA), nitrofurantoin (NIT), sulfonamides (sulfamethoxazole [SXT] and, sulfisoxazole [SXZ]), furazolidone (FZD), streptomycin (STR), trimethoprim (TMP), and tetracyclines (tetracycline [TET] and doxycycline [DOX]).

We inferred serogroups and serotypes of the *Vc* isolates using an in silico genomic-based approach based on mapping nucleotide sequence *k*-mers of each isolate against all known reference LPS O-antigen biosynthesis gene cluster sequences to determine the specific serogroups and serotypes of our isolates[55]. We inferred the serotype as the reference LPS O-antigen biosynthesis gene cluster with the highest sequence coverage than the rest of the LPS O-antigen sequences. We also compared the *wbeT* gene (GenBank accession: JF284685) against the O1 *Vc* isolates inferred to express the Inaba and Ogawa serotypes. We extracted the *wbeT* sequences from the Inaba and Ogawa *Vc* isolates using BLASTN (version 2.12.0 + )[95] and "getfasta" option implemented in bedtools (version 2.30.0)[96].

Lastly, we checked the presence and absence of MGEs, including pathogenicity islands, prophages, and ICEs. Due to the genomic variability of these sequences, driven by horizontal gene transfer, and their variably larger sizes compared to individual genes, we inferred their presence in the genomes based on *k*-mer sequences using KMA (version 1.4.12a)[105]. We specified the following flags when running KMA "-ef -dense -ex_mode -mct 1.0 -1t1 -mrs 0.1" to improve the query coverage so that the results were consistent with those based on visual inspection of the comparison of each genome and the MGEs using BLASTN (version 2.12.0 + )[95] and ACT (version 18.1.0)[106]. We mapped *k*-mers of each *Vc* isolate against reference SXT-like ICE sequences, namely, the original SXT (GenBank accession: KJ817376), ICEVchban5 (GenBank accession: GQ463140), ICEVchind4 (GenBank accession: GQ463141), ICEVchind5 (GenBank accession: GQ463142), ICEVchmex1 (GenBank accession: GQ463143), ICEVflInd1 (GenBank accession: GQ463144), ICEVchHai1 (JN648379), ICE$^{TET}$ (GenBank accession: MK165649), and ICE$^{GEN}$ (GenBank accession: MK165650). We repeated this analysis for other MGEs and pathogenicity islands, including PLE1 (GenBank accession: KC152960), PLE2 (GenBank accession: KC152961), VPI-1 (GenBank accession: AF325733), VPI-2 (GenBank accession: EU272902), ICP-1-like vibriophage (GenBank accession: MN402506), CTXφ prophage (GenBank accession: KJ540278), and VSP-II (GenBank accession: KM660639). We used minimum cut-offs of 40% and 80% for the query sequence coverage and identity, respectively. Considering the high genetic diversity of the CTXφ prophage, we further confirmed its presence by mapping the reads

of each *Vc* isolate against the sequence of an intact CTXφ prophage (GenBank accession: KJ540278) using Snippy (version 4.6.0) (https://github.com/tseemann/snippy).

## Reporting summary

Further information on research design is available in the Nature Portfolio Reporting Summary linked to this article.

## Data availability

We have deposited the whole-genome sequence data for the study isolates in the National Center for Biotechnology Information (NCBI) Sequence Read Archive (BioProject number: PRJNA974496). The accession numbers and other metadata for our isolates are available in the Supplementary Data 1 file. All the other data supporting the findings of this study are described in this paper or are available as part of the supplementary material.

## Code availability

We have described all the tools and methods used for the analysis in the Material and Methods sections.

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

## Acknowledgements
We acknowledge the support of the Ministry of Health and clinical staff in Malawi, who cared for the patients presenting with cholera-like diarrhoea and collected the samples used in this study. We thank the National Microbiology Reference Laboratory (NMRL) at the Public Health Institute of Malawi (PHIM) for coordinating sample collection and compiling data on the number of cholera cases and deaths in Malawi. We also thank the Next Generation Sequencing Unit and Division of Virology at the University of the Free State for timely genome sequencing services and technical support, especially Mil-ton Mogotsi and the wet-lab team. We are also grateful to Dr Florent Lassalle at the Wellcome Sanger Institute for providing advice regarding genomic-based serotyping of *Vc* and kindly sharing a reference dataset of LPS O-antigen loci for all known *Vc* serogroups. Finally, we thank the Yale Center for Research Computing at Yale University and Wellcome Sanger Institute for providing computa-tional resources used for the bioinformatic analysis. This research was funded by the UK National Institute for Health and Care Research (NIHR) Global Health Research Group on Gastrointestinal Infections (grant number: NIHR133066) using UK Aid from the UK Government to support global health research; and the Bill and Melinda Gates Foundation (Grant number: Investment number: INV-046917). N.A.C. is an NIHR Senior Investigator (NIHR203756). N.A.C., D.H., and K.C.J. are affiliated with the NIHR Health Protection Research Unit in Gas-trointestinal Infections at the University of Liverpool, a partnership with the UK Health Security Agency (UKHSA), in collaboration with the University of Warwick. The funders had no role in the study design, data collection, and interpretation, or the decision to submit the work for publication. The authors did not receive any financial support or other forms of reward related to the development of the manuscript. Therefore, the findings and conclusions in this report are those of the authors and do not necessarily represent the formal position of the funders. The views expressed are those of the author(s) and not necessarily those of the NIHR, the Department of Health and Social Care or the UK Health Security Agency or the UK government.

## Author contributions
C.C., I.C., W.K., N.A.C., and K.C.J., conceived and designed the study. I.C., W.K., and K.C.J. performed sample selection. I.C. and W.K. per-formed microbiological work. M.M.N. carried out whole-genome sequencing of the samples. K.C.J., I.C., B.M., C.M., Q.D., and U.L.M. assisted with compiling cholera case and death data. C.C., I.C., and K.C.J. analysed and interpreted the data. S.N. and D.K. assisted with data management and analysis. C.C. and K.C.J. wrote the initial draft of the paper. C.C., I.C., P.M., W.K., D.C., C.M., U.L.M., J.B-B, K.C.Jambo, B.M., W.Kapindula, P.B., R.J.M, S.N., D.K., C.A.M., A.K., L.N., A.D.S., A.C-M., A.W.K., M.M.N., D.H., N.F., M.K., Q.D., C.L.M., N.A.C., and K.C.J. reviewed and approved the manuscript.

## Competing interests
The authors declare no competing interests.

## Additional information

[1]Department of Epidemiology of Microbial Diseases, Yale School of Public Health, Yale University, New Haven, CT, USA. [2]Yale Institute for Global Health, Yale University, New Haven, CT, USA. [3]Department of Clinical Infection, Microbiology and Immunology, Institute of Infection, Veterinary and Ecological Sciences, University of Liverpool, Liverpool, UK. [4]NIHR Mucosal Pathogens Research Unit, Research Department of Infection, Division of Infection and Immunity, University College London, London, UK. [5]Parasites and Microbes Programme, Wellcome Sanger Institute, Hinxton, UK. [6]Public Health Institute of Malawi, Ministry of Health, Lilongwe, Malawi. [7]Department of Pathology, School of Medicine and Oral Health, Kamuzu University of Health Sciences, Blantyre, Malawi. [8]Malawi-Liverpool-Wellcome Research Programme, Blantyre, Malawi. [9]Department of Medical Laboratory Sciences, Faculty of Biomedical Sciences and Health profession, Kamuzu University of Health Sciences, Blantyre, Malawi. [10]Ministry of Health, Balaka District Hospital, Balaka, Machinga, Malawi. [11]Department of Psychiatry, University of British Columbia, Vancouver, BC, Canada. [12]Department of Clinical Sciences, Liverpool School of Tropical Medicine, Liverpool, UK. [13]Department of Biomedical Sciences, School of Life Sciences and Allied Health Professions, Kamuzu University of Health Sciences, Blantyre, Malawi. [14]Diarrhoeal Pathogens Research Unit, Sefako Makgatho Health Sciences University, Medunsa 0204 Pretoria, South Africa. [15]NIHR Health Protection Research Unit in Gastrointestinal Infections, University of Liverpool, Liverpool, UK. [16]Next Generation Sequencing Unit and Division of Virology, Faculty of Health Sciences, University of the Free State, Bloemfontein 9300, South Africa. [17]Malawi Ministry of Health, Lilongwe, Malawi. [18]NIHR Global Health Research Group on Gastrointestinal Infections, University of Liverpool, Liverpool, UK. [19]These authors contributed equally: Chrispin Chaguza, Innocent Chibwe. [20]These authors jointly supervised this work: Chisomo L. Msefula, Nigel A. Cunliffe, and Khuzwayo C. Jere. ✉e-mail: chrispin.chaguza@yale.edu; Khuzwayo.Jere@liverpool.ac.uk

