## [Peer Review File · Nature Communications]

Genomic insights into the 2022–2023 *Vibrio cholerae* outbreak in MalawiREVIEWER COMMENTS

Reviewer #1 (Remarks to the Author):

This is a timely genomic analysis of a large, still ongoing cholera outbreak in Malawi. Chaguzo et al. determined the genome sequences from 45 *Vibrio cholerae* strains collected in Malawi and compared them to previously published data from international sources. Their main conclusion is that the current outbreak is caused by a *V. cholerae* strain that has emerged somewhere in Asia and then spread internationally during the last two decades.

The reported analysis is very useful as it can guide public health measures to curb the outbreak. However, the introduction of this particular *V. cholerae* lineage into Africa has been reported some years ago (lineage 'T12', Weill et al., ref. 7).

The repeated statements that the causative strain originated in Asia may be misleading. Such claims on the geographic origins of an epidemic may cause resentment against specific ethnic groups and hence should be made with utmost caution. While it is true that the strain may have emerged somewhere in Asia many years ago, it has since become endemic in numerous countries in Africa and has also caused outbreaks in several countries in Central America. Therefore, it is quite unlikely that the strain was introduced into Malawi directly from Asia.

For example, in the Abstract, it is claimed that "the Malawi outbreak strains originated from Asia". Instead, it is more likely that the strain was introduced from a neighboring country in Africa, where T12 is endemic. It is also possible that it was introduced from Mexico or from some other country outside of Asia.

The paper is a bit long and redundant in some places. For example, in the Discussion section, antimicrobial resistance (AMR) traits are discussed twice, first in lines 527 ff, and then again from line 592. In addition, there are three pages on AMR in the Results section, which could be shortened.

The reporting of virulence genes is excessive, too. Since the predominant outbreak strain is closely related to *V. cholerae* isolates that were previously reported from Haiti and elsewhere, a similar endowment with virulence genes can be expected.

The Supplementary Material is confusing. There are 11 Tables with distributed information on the same 51 bacterial isolates; is all that information on gene presence/absence relevant to the reader? It would probably be helpful to reduce the number of Suppl. Tables. Further, it is unclear (to this Reviewer), why there are 30 pages with 28 variants of Supplementary Figure 2, all with identical figure captions, but displaying different curve extensions.

Reviewer #2 (Remarks to the Author):

The authors present here an epidemiological description of the ongoing cholera outbreak in Malawi as well as presenting finding from whole genome sequencing of *Vibrio cholerae* strains from the outbreak. The current cholera outbreak is the largest in recent history and the description of epi and genomic properties of the strains is important to elucidate. This manuscript is led by Malawians and the Malawian MOH is directly involved. This stands in contrast to many of the other cholera genomics and epi papers of late.

I have specific comments, but upfront there are three things that need addressing. The first is that there is something that has gone wrong in the whole genome and subsequent SNP alignment. The numbers being reported are way out of line (as in they are too small) from what we normally see from WGS data from cholera outbreaks. Maybe it was the overly aggressive trimming of the alignment with trimAI? Most folks have been using snp-sites to generate the SNP alignment. The samples over the course of the outbreak should definitely have some variation. We can even

distinguish households from each other by genomics in other settings during outbreaks. In other studies, we see somewhere on the order of ~10,000 SNPs for the 7PET total dataset, which includes strains from Wave 1 through to current outbreaks.

Most other studies have mapped the short-read data against the reference to call SNPs rather than using assemblies. I can probably guess that the rationale behind using assemblies was the PathogenWatch database contains the assemblies, but something has definitely gone a bit wonky. I would encourage the authors to run snippy using the short read data. You can then directly merge your alignment with the large one used in recent publications that was also produced via the snippy pipeline (<https://figshare.com/s/b70a9efac9cf2625480e>). This alignment doesn't have newer samples included but should get the authors off to a good start.

Basic workflow for 7PET isolates:

Map with snippy - combine alignment - Gubbins - snp-sites for SNP alignment

The second main thing is that I feel there was a bit of a missed opportunity here with the analyses. This likely could have been driven by the fact that the alignment showed zero SNPs between the outbreak strains, so I don't want to be too harsh. I'm hoping that with the alignment issue solved, the authors could dig a bit deeper into if there is any geographic clustering, and within country transmission analyses etc. Most of the analysis in the paper centers around comparisons of these current outbreak strains to the limited number of older strains from Malawi the 1990s. There wasn't much discussion / analyses about how these strains compare to other currently circulating strains in Africa or the region.

I suppose the elephant in the room is this recent EID paper with travel related cases from recent Malawi and South African cholera WGS data (https://wwwnc.cdc.gov/eid/article/29/8/23-0750_article). This paper only included 2 genomes from Malawi and 4 from South Africa – but it appears it is likely the same strain you are reporting on. They found this to be a novel introduction event (AFR 15). This is where your deeper sampling can provide additional, deeper insight on within country transmission etc. Adding these samples and the recent Pakistan samples to the alignment should more fully contextualize the 45 7PET strains in your study. Further to the point about your analyses showing 0 SNPs, in the EID paper, they found pairwise SNP distances between these 6 isolates ranging from 0-6 SNPs.

I have some additional comments below, but for many of my others I will hold off until the authors sort out the SNP alignment – as I think this will change things dramatically.

Line 96-98: This is not necessarily true. That manuscript documented at least 12 introductions (now up to 15), into Africa since 1970 with the most recent ones being MDR. They were all distinct sub-lineages of 7PET.

Line 102-108: I believe you have got it all right, but clean up for clarity. Make sentences shorter and be clear about what factors / pathogenicity islands are associated with which biotypes.

Line 131: Delete "specifically"

Line 195 section: I was struck a bit that you had such low recovery of Vc from the presumptive positive samples. Do you still have access to the original samples to try a PCR for Vc? Maybe it was just bad luck on the colonies picked from the plate?

Line 215-217: Shouldn't TCBS agar be pretty differential against those organisms you found?

Line 237: For the Inaba strains, were you able to look at the wbeT gene to detect mutations / deletions?

Line 257: The 93,704 SNPs has to be inclusive of the non-7PET strains? Interesting that you found some non-O1s. For those diverse non-O1 strains, it is probably better to use assemblies and run a pan-genome tool, like Roary or Panaroo, and then use the core genes for the alignment for a more "species wide tree".

Line 267: That 20 SNPs between the strains that are placed basal and to the others is one of the major red flags. The numbers just don't make sense with any of the other data produced over the years. This number should at least be in the hundreds if the placement of the strains is correct.

Lines 276-291: Numbers will almost certainly change, and will likely demonstrate definitively that the current outbreak strain is not directly related to the earlier 1990s strains.

Line 306: I don't think we have enough sampling outside of 7PET to have any confidence about historical introductions and relatedness of these non-O1 strains. Would remove the bit about possible introduction.

Lines 309-345: As stated above, much of this is likely to change and needs to incorporate the EID article describing the AFR-15 (T15) introduction, which is likely what these strains are as well. Thanks for providing a Microreact site – it is always a nice way to interactively view the phylogeny and data.

Line 360-364: Strains sequenced from those outbreaks found nothing of note that would explain higher virulence - but rather the humanitarian crises in these settings provided the opportunity for cholera. This was critically highlighted for Yemen and Haiti – as it is important to not shift the conversation away from ultimate root causes that can potentiate cholera outbreaks. Yes ctxB7 has been implicated as potentially producing more cholera toxin, but many strains circulating since 2004 have ctxB7 and they have wildly different attack rates and CFRs across outbreaks.

Lines 369-376: I'm a bit unsure why this analysis of the presence of VC2346 is included under the 'virulence' paragraph. To my knowledge VC2346 isn't a known virulence gene, but rather a marker for 7PET.

Line 385 – 387: You might check via mapping the reads to see if you really have a loss of CTX. Although, we have seen spontaneous excision of CTX in several datasets – so not unfeasible for this to occur.

Lines 544-549: I think this is conjecture and probably a bit of an overreach. I'd recommend taking this out.

Lines 630-638: I'm hoping that the authors will be able to carry out some of these analyses now with the alignment sorted out.

Line 702: Presumably these samples were all positive on the rapid kit?

Line 703: This line struck me. Why were samples only collected for patients with known antibiotic consumption?

Line 745-763. Much as already been discussed about this. In recent studies we have been rooting the 7PET phylogeny on strain A6, and not including any pre-7PET or non7PET strains so you can actually get the resolution within the phylogenetic tree. If you don't kick those strains out, the 7PET just looks like a line and you can't see the branching patterns within the 7PET lineage.

I'm very much looking forward to the next iteration of this manuscript.

Reviewer #3 (Remarks to the Author):

Review Report:

This study has a good potential for the understanding of the cholera epidemics in Malawi especially in the climate change context. However the work conducted here needs substantial improvements. Abstract: Lines 60-62 in the abstract do not provide clear understanding because these features described here are already known in the T10 and T13 7PET strains in the region. They could be

new, but they don't seem so.

Introduction: Lines 96-98 seem to be limited to T12 strains while ignoring recent literature that indicates the presence of T13 strains in East Africa, particularly in Tanzania, Kenya, and Uganda since 2017. Additionally, other T13 strains have been found in Zimbabwe in 2018, exhibiting newly acquired resistance profiles including an ESBL gene. Overall, the literature search needs to be more comprehensive without ignoring already known facts.

Virulence Factors: The *ctxB* genotype should not be regarded as a virulence factor but rather as a biotyping marker. The focus should be on *ctxA*, which plays a crucial role in cholera virulence, as well as the TCPs and the other genes on the prophage as well as the VSPs. So far, no new findings have been reported regarding the virulence profile of these isolates, so this information could be summarized in a single sentence. Furthermore, it should be considered that isolates lacking the *ctxA/B* complex may not be regarded as toxigenic, unless it is a sequencing issue, which should be discussed further.

Also, It would be valuable to confirm the presence of Ogawa and Inaba serotypes through genomic analysis of the *wbE* gene.

Global Phylogeny: The global phylogeny analysis is missing key guides to demonstrate the sublineages of these isolates. The main isolate of the Malawi outbreaks appear to belong to the T13 lineage. Considering the importance of this study and its implications for future reference in the global phylogeny of Malawi isolates, it is necessary to rerun the phylogenetic analysis. This analysis should include annotating clearly on the tree, all 7PET sublineages from T1 to T13, incorporating the latest T13 strains from Zimbabwe in 2018 (New England Journal) and East Africa (refer to EID and PLOS NTD papers from 2019/2023). Rooting the tree on the reference N16961 strain is essential.

It is important for the authors to confirm whether the previous Malawi strains are T10 in their phylogeny analysis and report it accordingly to ensure universality in the field.

Results and Discussion: Instances where the results are discussed or compared to other studies should be moved to the discussion section, as per the designated structure of the report.

Environmental Isolate: The inclusion of a single environmental isolate is insufficient to draw conclusions regarding clinical versus environmental aspects.

Resistance Profile: The resistance profile should be verified for comparison with other T10 and T13 strains, as this information is already known both for the genotypes and the MGEs. So these should be reported not as new findings but be put in existing context. None of these ICs found are new.

In conclusion it seems like the Malawi isolates are likely just a continuation of the existing sub lineages in the region, now a new introduction. Otherwise, authors need to argue for it more effectively with their data.

Overall, to enhance the comprehensiveness of the report, it is necessary to address the mentioned points and incorporate the appropriate revisions and reorganizations.

Reviewer #1 (Remarks to the Author):

This is a timely genomic analysis of a large, still ongoing cholera outbreak in Malawi. Chaguza et al. determined the genome sequences from 45 *Vibrio cholerae* strains collected in Malawi and compared them to previously published data from international sources. Their main conclusion is that the current outbreak is caused by a *V. cholerae* strain that has emerged somewhere in Asia and then spread internationally during the last two decades.

Response: Thank you for the excellent summary of the approaches and main findings of the paper.

The reported analysis is very useful as it can guide public health measures to curb the outbreak. However, the introduction of this particular *V. cholerae* lineage into Africa has been reported some years ago (lineage 'T12', Weill et al., ref. 7).

Response: We agree that our findings can guide public health measures to curb the current and future outbreaks. Indeed, the T12 *V. cholerae* lineage has been reported by Weill et al. (PMID: 29123067). After further analysis as suggested by reviewer #2, including additional recent contextual genomes from Asia, we have revised the text to correct that the 2022 Malawi strains represent a new lineage, known as T15 or AFR15, rather than T12 as we previously reported. Based on this, our findings are now consistent with those from a recent study from South Africa based on imported *V. cholerae* strains into South Africa from Malawi (PMID: 37352549) and another recent preprint focusing on the strains from Malawi (<https://www.medrxiv.org/content/10.1101/2023.08.22.23294324v1>).

The repeated statements that the causative strain originated in Asia may be misleading. Such claims on the geographic origins of an epidemic may cause resentment against specific ethnic groups and hence should be made with utmost caution. While it is true that the strain may have emerged somewhere in Asia many years ago, it has since become endemic in numerous countries in Africa and has also caused outbreaks in several countries in Central America. Therefore, it is quite unlikely that the strain was introduced into Malawi directly from Asia.

Response: Thank you for pointing this out. We have revised the statement regarding the origin of the Malawi 2020 cholera outbreak strains in Reprahsed the abstract from

“...suggested the Malawi outbreak strains originated from Asia” to “Phylogenetic reconstruction revealed that the Malawi outbreak strains correspond to a recent importation from Asia into Africa (sublineage AFR15)”. Indeed, the strains may have come from other countries and not been introduced into Malawi directly. This statement is consistent with the findings from recent work by Smith et al (PMID: 37352549) based on locally transmitted South African strains and imported strains from Malawi. We believe rephrasing the sentences this way will improve clarity and avoid misleading interpretations of the origin of the strains.

For example, in the Abstract, it is claimed that "the Malawi outbreak strains originated from Asia". Instead, it is more likely that the strain was introduced from a neighboring country in Africa, where T12 is endemic. It is also possible that it was introduced from Mexico or from some other country outside of Asia.

Response: We have revised the sentence regarding the origin of the *V. cholerae* strains. Please see the previous comment before this one. As stated in the earlier comments, we have revised the analysis to include recent genomes, mostly from Asia. Consistent with recent work in South Africa (PMID: 37352549) and Malawi (<https://www.medrxiv.org/content/10.1101/2023.08.22.23294324v1>), it appears that the imported lineage is not T12 but T15 or AFR15. We have revised the manuscript to reflect this.

The paper is a bit long and redundant in some places. For example, in the Discussion section, antimicrobial resistance (AMR) traits are discussed twice, first in lines 527 ff, and then again from line 592. In addition, there are three pages on AMR in the Results section, which could be shortened.

Response: We have merged the discussion of the AMR in the discussion and shortened some paragraphs where possible.

The reporting of virulence genes is excessive, too. Since the predominant outbreak strain is closely related to *V. cholerae* isolates that were previously reported from Haiti and elsewhere, a similar endowment with virulence genes can be expected.

Response: Thank you for the suggestion. We have cut down on the discussion of the virulence factors to mostly focus on those associated with the CTX prophage, VPI, and VSPs, and referenced the papers describing the 7PET strains associated with outbreaks in Haiti, Pakistan and other settings; including PMID: 30602788 and PMID: 21142692.

We have also provided further information for the distribution of the virulence genes in Supplementary Data 2.

The Supplementary Material is confusing. There are 11 Tables with distributed information on the same 51 bacterial isolates; is all that information on gene presence/absence relevant to the reader? It would probably be helpful to reduce the number of Suppl. Tables. Further, it is unclear (to this Reviewer), why there are 30 pages with 28 variants of Supplementary Figure 2, all with identical figure captions, but displaying different curve extensions.

Response: Thank you for pointing this out. We used different tools and databases for the analyses of virulence, AMR, and other genes. As suggested, we have merged the previous Supplementary Data tables 2-11 into the supplementary into a new single spreadsheet Supplementary Data 2.

Regarding Supplementary Figure 2, each figure represents the number of observed cases and deaths and the case fatality ratio in each of the twenty-eight districts in Malawi. We have now revised the previous Supplementary Figure 2 to separate it into multiple Supplementary Figures, each one showing data for each district in Malawi during the 2022–2023 cholera outbreak.

Reviewer #2 (Remarks to the Author):

The authors present here an epidemiological description of the ongoing cholera outbreak in Malawi as well as presenting findings from whole genome sequencing of *Vibrio cholerae* strains from the outbreak. The current cholera outbreak is the largest in recent history and the description of epi and genomic properties of the strains is important to elucidate. This manuscript is led by Malawians and the Malawian MOH is directly involved. This stands in contrast to many of the other cholera genomics and epi papers of late.

Response: Thank you for the excellent summary of the paper. We have addressed the specific comments below.

I have specific comments, but upfront there are three things that need addressing. The first is that there is something that has gone wrong in the whole genome and

subsequent SNP alignment. The numbers being reported are way out of line (as in they are too small) from what we normally see from WGS data from cholera outbreaks. Maybe it was the overly aggressive trimming of the alignment with trimAl? Most folks have been using snp-sites to generate the SNP alignment. The samples over the course of the outbreak should definitely have some variation. We can even distinguish households from each other by genomics in other settings during outbreaks. In other studies, we see somewhere on the order of ~10,000 SNPs for the 7PET total dataset, which includes strains from Wave 1 through to current outbreaks.

Response: Thank you for noticing that something may have gone wrong with the alignment. This is indeed correct. It turns out that we did an aggressive trimming of the alignment using trimAl. We also used snp-sites followed by trimAl. However, based on your suggestions, we decided to exclude the trimAl step and directly use snp-sites in the whole-genome alignment of the strains. Indeed, the new alignment of all the *Vc* sequences (7PET and non-7PET) contained ~6,240 SNPs (excluding the outgroup species), while the alignment of only 7PET sequences had ~14,557 SNPs (~20,993 before removing recombinogenic regions), which is consistent with the number of SNPs mentioned in the comment above. These SNPs are consistent with those reported in other studies (after adjusting for the number of isolates included in the alignment), for example, PMID: 37770747 (37,170 SNPs based on fewer genomes from a single setting only), PMID: 29123068 (9,300 SNPs for 7PET; 115,789 SNPs for non-7PET alignment), and PMID: 36859426 (6,399 SNPs based on fewer 7PET isolates from a single country only). We are confident that the revised analysis is more robust and based on correct alignment, which does not underestimate the diversity, especially between the 7PET *Vc* strains.

Most other studies have mapped the short-read data against the reference to call SNPs rather than using assemblies. I can probably guess that the rationale behind using assemblies was the PathogenWatch database contains the assemblies, but something has definitely gone a bit wonky. I would encourage the authors to run snippy using the short read data. You can then directly merge your alignment with the large one used in recent publications that was also produced via the snippy pipeline (<https://figshare.com/s/b70a9efac9cf2625480e>). This alignment doesn't have newer samples included but should get the authors off to a good start.

Response: We used reads, where they were available, and assemblies obtained from PathogenWatch. As mentioned in the response to the previous comment, we have revised the analysis based on a new whole-genome alignment. We removed the alignment trimming step that used trimAl as it removed sites that shouldn't have been removed in the alignment, resulting in fewer than expected SNPs. We also added

contextual sequences from recent outbreaks, such as in 2022 in Pakistan and other places (PMID: 37352549), for a more robust phylogeny.

Basic workflow for 7PET isolates:

Map with snippy - combine alignment - Gubbins - snp-sites for SNP alignment

Response: We agree with this basic workflow for analysing 7PET strains. We have repeated the analysis based on this approach to generate the phylogeny of the 7PET isolates from Malawi in the context of 7PET sequences from other countries globally.

The second main thing is that I feel there was a bit of a missed opportunity here with the analyses. This likely could have been driven by the fact that the alignment showed zero SNPs between the outbreak strains, so I don't want to be too harsh. I'm hoping that with the alignment issue solved, the authors could dig a bit deeper into if there is any geographic clustering, and within country transmission analyses etc. Most of the analysis in the paper centers around comparisons of these current outbreak strains to the limited number of older strains from Malawi the 1990s. There wasn't much discussion / analyses about how these strains compare to other currently circulating strains in Africa or the region.

Response: Indeed, the zero SNP differences were due to the previous issues with the whole-genome alignment, as addressed in the previous comments. We have now revised the estimates for the number of SNPs between the isolates and the results are as expected, that there is some diversity between the outbreak *Vc* strains during the 2022-2023 cholera outbreak in Malawi.

Unfortunately, due to the sampling of isolates mostly from the central region of Malawi, we did not have enough resolution to do detailed geographic and within-country transmission analyses. We would like to note that there is no robust routine surveillance system for diseases like cholera, especially at the molecular level, which limits the availability of samples for this study. Essentially, we sequenced the available samples, which, of course, had limitations in terms of being few and not geographically or temporally representative. However, we agree that such analyses could have provided crucial insights regarding the spread of *Vc* strains in Malawi. We have noted this as a limitation of the study in the discussion. We want to use this study to showcase that there is value in routine surveillance of *Vc* strains across the country to address these knowledge gaps.

We also agree that we mostly compared the current outbreak and the historical strains collected from Malawi. We have added some results section titled "*Genetic analysis*

suggests the 2022–2023 outbreak-associated V. cholera isolates were recently imported from Asia” on the similarity of the Malawi outbreak strains and those from other African countries.

I suppose the elephant in the room is this recent EID paper with travel related cases from recent Malawi and South African cholera WGS data (https://wwwnc.cdc.gov/eid/article/29/8/23-0750_article). This paper only included 2 genomes from Malawi and 4 from South Africa – but it appears it is likely the same strain you are reporting on. They found this to be **a novel introduction event (AFR 15)**. This is where your deeper sampling can provide additional, deeper insight on within country transmission etc. Adding these samples and the recent Pakistan samples to the alignment should more fully contextualize the 45 7PET strains in your study. Further to the point about your analyses showing 0 SNPs, in the EID paper, they found pairwise SNP distances between these 6 isolates ranging from 0-6 SNPs.

Response: Thank you for pointing this out. We agree that it’s critical to include the Pakistan genomes and genomes from imported cases in South Africa reported by Smith et al (PMID: 37352549). As suggested, we included these genomes in our analysis and generated a correct whole genome alignment based on the approach suggested in the earlier comments. Indeed, our results show that there was a novel introduction to Malawi, which was termed AFR15 or T15 by Smith et al (PMID: 37352549). We have revised the text to show the correct number of SNPs between the isolates based on the correct alignments and mention the novel introduction event (AFR15) consistent with the report from South Africa. As noted in the previous comments, unfortunately, we are unable to do a more detailed within-country analysis as we did not have access to samples collected across the whole country.

I have some additional comments below, but for many of my others I will hold off until the authors sort out the SNP alignment – as I think this will change things dramatically.

Response: We have corrected the alignment as suggested in the earlier comments, and the revised findings are now more robust and consistent with what may have been expected. Thank you for allowing us to correct the results rather than completely dismissing the previous findings based on the incorrect alignment.

Line 96-98: This is not necessarily true. That manuscript documented at least 12 introductions (now up to 15), into Africa since 1970 with the most recent ones being MDR. They were all distinct sub-lineages of 7PET.

Response: We agree with this comment and have now corrected the sentence as suggested: “Detailed phylogeographic analysis revealed fifteen independent introductions of *Vc* into Africa from other continents (designated T1-T15) due to both antibiotic-susceptible and multidrug-resistant (MDR) *Vc* lineages from 1970s^{7,17,22,23}”.

Line 102-108: I believe you have got it all right, but clean up for clarity. Make sentences shorter and be clear about what factors / pathogenicity islands are associated with which biotypes.

Response: Thank you for the suggestion. We have revised the sentences to make them shorter for clarity and mentioned that the mentioned virulence factors are mostly associated with the El Tor biotype strains responsible for the 7th pandemic. Here is the revised sentence “We first analysed the distribution of the *ctxA* gene, which encodes a cholera toxin (CT) and is found on the filamentous lysogenic CTX ϕ prophage^{24,25}. We found that the majority of the clinical *Vc* isolates carried the *ctxA* virulence gene. However, this gene may have been missed in some of the *Vc* isolates as all the 2022–2023 Malawi cholera outbreak *Vc* isolates carried the CTX ϕ prophage (Supplementary Data 4). Interestingly, a single environmental 7PET ST69 *Vc* isolate sequenced in the present study showed the absence of *ctxA* gene although it clustered with the rest of the outbreak-associated 7PET *Vc* isolates (Fig. 4a). We also found a similar distribution of additional CTX ϕ prophage core genes considered to play a role in *Vc* virulence and pathogenicity in the outbreak-associated and historical 7PET isolates, but not in the non-7PET isolates widely associated with sporadic cholera cases. Besides the CTX ϕ prophage, all the 2022–2023 outbreak-associated and historical 7PET isolates harboured the *Vc* pathogenicity island 1 (VPI-1), which harbours the second most important *Vc* virulence factor²⁸ (Fig. 4a, Supplementary Data 2).”.

Line 131: Delete “specifically”

Response: We have deleted this word from the sentence.

Line 195 section: I was struck a bit that you had such low recovery of *Vc* from the presumptive positive samples. Do you still have access to the original samples to try a PCR for *Vc*? Maybe it was just bad luck on the colonies picked from the plate?

Response: We were equally surprised by the slightly lower recovery of *Vc* from the presumptive positive samples. Unfortunately, we are unable to access the original

samples as they were processed by a government laboratory, which does not routinely collect and store the samples. However, we agree that performing PCR for *Vc* on the samples could have provided a more robust estimate of the proportion of the samples that were indeed positive for *Vc*. We have included a detailed statement on this as a limitation in the discussion, as it is possible that these findings may merely reflect the lack of the availability of a full battery of tests used to distinguish *Vc* from the other pathogen or other issues with the culturing of the isolates. We also noted that further studies are needed to determine the prevalence of non-*Vc* bacterial pathogens in samples from patients with cholera-like diseases, especially in Africa.

Line 215-217: Shouldn't TCBS agar be pretty differential against those organisms you found?

Response: Yes, the TCBS is selective for *Vibrio*, but it supports growth of several other bacterial genera that can, of course, be differentiated from *Vibrio* due to their distinctive green phenotypes - similar to those of *V. vulnificus*, for instance, *Pseudomonas*, *Aeromonas*, *Pseudoalteromonas*, *Shewanella* and *Staphylococcus*, *Flavobacterium*, (PMID: 15353563). Others have used modified TCBS, which is highly specific and we did not have (PMID: 31318499) or further performed follow-up testing to confirm green colonies as *V. vulnificus*, for instance, Colistin-polymyxin B-cellobiose (CPC) agar (<https://www.sciencedirect.com/science/article/abs/pii/B9780123971692000664>). We did not observe green colonies, so no follow-up tests were required, but there is a possibility that some of the colonies we picked to extract DNA were mixed.

Line 237: For the Inaba strains, were you able to look at the *wbeT* gene to detect mutations / deletions?

Response: We have now compared the *wbeT* gene of the Inaba and Ogawa strains. We extracted the *wbeT* sequences from the Inaba and Ogawa *Vc* isolates using BLASTN (PMID: 2231712) and "bedtools getfasta" (PMID: 20110278). However, we did not find any specific mutations distinguishing these serotypes (Supplementary Data 3). These findings are consistent with findings from a study of serotype switching in Haiti which also reported that the serotype Inaba emergence was not linked to specific mutations in the *wbeT* gene (PMID: 36537825). We have updated the methods section and results section accordingly.

Line 257: The 93,704 SNPs has to be inclusive of the non-7PET strains? Interesting that you found some non-O1s. For those diverse non-O1 strains, it is probably better to

use assemblies and run a pan-genome tool, like Roary or Panaroo, and then use the core genes for the alignment for a more “species wide tree”.

Response: Indeed, the 93,704 SNPs included non-7PET strains while the alignment without the outgroup *V. mimicus* genome contained 76,240 SNPs. When looking at the core SNPs, using the Panaroo would generate similar results (same for Roary to some extent, although it was shown that it overestimates the pan-genome size [PMID: 32698896]), as the additional genes constituting the accessory genome, therefore, absent in the reference genome would be excluded in the Panaroo alignment.

Line 267: That 20 SNPs between the strains that are placed basal and to the others is one of the major red flags. The numbers just don’t make sense with any of the other data produced over the years. This number should at least be in the hundreds if the placement of the strains is correct.

Response: We agree with this assessment. The number of SNPs was based on the aggressively trimmed alignment and was indeed incorrect. As mentioned in the earlier comments, we have now dropped the alignment trimming step using trimAl, and the new number of SNPs show a higher diversity between the strains placed at the base of the phylogeny and the rest of the Malawi 2022-2023 outbreak *Vc* strains.

Lines 276-291: Numbers will almost certainly change, and will likely demonstrate definitively that the current outbreak strain is not directly related to the earlier 1990s strains.

Response: Indeed, as mentioned in the previous comment, our revised results show definitively that the 2022-2023 outbreak *Vc* strains are not directly related to the earlier 1990s strains in Malawi. The 2022-2023 Malawi outbreak strains are closest to the 2022 Pakistan outbreak strains. We added the Pakistan strains to the phylogeny when revising the manuscript, as suggested in the earlier comments.

Line 306: I don’t think we have enough sampling outside of 7PET to have any confidence about historical introductions and relatedness of these non-O1 strains. Would remove the bit about possible introduction.

Response: Indeed, we cannot say anything about the potential introduction of non-O1 strains. As suggested, we have removed the sentence about possible introduction events in the sentence. **“The non-ST69 or non-7PET Malawian isolates deemed to cause sporadic cases during the 2022–2023 outbreak, belonging to ST40 and ST635**

clones, clustered closest in the global phylogeny with a single isolate from India and three isolates from Austria of identical STs, respectively (Fig. 3). The single ST40 isolate from India was collected in 1962 and differed from the 2023 Malawi isolate by ~596 SNPs, while ~5,299 SNPs separated the 2022 Malawi ST635 isolate from the three recent Austrian ST635 isolates sampled in 2012.”

Lines 309-345: As stated above, much of this is likely to change and needs to incorporate the EID article describing the AFR15 (T15) introduction, which is likely what these strains are as well. Thanks for providing a Microreact site – it is always a nice way to interactively view the phylogeny and data.

Response: Indeed, based on the correct alignment, which included the 2022 Pakistan strains, we have revised the manuscript to describe the AFR15 (T15) introduction. These findings are now consistent with the reports from South Africa and Malawi. Thank you for highlighting the usefulness of interactively viewing the phylogeny and strain metadata. We have also updated the Microreact phylogeny and data.

Line 360-364: Strains sequenced from those outbreaks found nothing of note that would explain higher virulence - but rather the humanitarian crises in these settings provided the opportunity for cholera. This was critically highlighted for Yemen and Haiti – as it is important to not shift the conversation away from ultimate root causes that can potentiate cholera outbreaks. Yes ctxB7 has been implicated as potentially producing more cholera toxin, but many strains circulating since 2004 have ctxB7 and they have wildly different attack rates and CFRs across outbreaks.

Response: Thank you for pointing this out. We have revised relevant sections of the manuscript to highlight the importance of the humanitarian crises and natural disasters in facilitating cholera outbreaks in settings such as Malawi. Here is the text we have added to the discussion to emphasize this **“Collectively, these findings emphasise the critical role of natural disasters, such as earthquakes and cyclones, and humanitarian crises, such as wars, which displaces individuals and disrupts water supplies and sanitation systems, in potentiating cholera outbreaks, even more so than the bacterial factors themselves. This highlights the need for investing in quality water supply and sanitation systems and other measures to minimise the occurrence of humanitarian crises, preparedness for environmental disasters, and the availability of oral cholera vaccines to reduce the risk of cholera outbreaks.”**

Lines 369-376: I'm a bit unsure why this analysis of the presence of VC2346 is included under the 'virulence' paragraph. To my knowledge VC2346 isn't a known virulence gene, but rather a maker for 7PET.

Response: We agree with this suggestion. We have moved the paragraph discussing VC2346 to the paragraph describing the biotypes and serotypes.

Line 385 – 387: You might check via mapping the reads to see if you really have a loss of CTX. Although, we have seen spontaneous excision of CTX in several datasets – so not unfeasible for this to occur.

Response: As suggested, we mapped the reads against the reference CTX sequence (GenBank accession: KJ540278) using snippy (<https://github.com/tseemann/snippy>). Based on this analysis, we found that all the 2022–2023 Malawi genomes contained the CTX phage. We have updated the results to mention that the isolates that did not appear to harbour the CTX phage (possibly reflecting the high genetic diversity of this MGE) actually contained a shorter version of this element based on the mapping analysis (Supplementary Data 4).

Lines 544-549: I think this is conjecture and probably a bit of an overreach. I'd recommend taking this out.

Response: As suggested, we have deleted this text from the manuscript.

Lines 630-638: I'm hoping that the authors will be able to carry out some of these analyses now with the alignment sorted out.

Response: Indeed, we have now done some of these analyses using the corrected whole genome alignment. We have not done phylogeographic and molecular dating analyses, as this has already been described in greater detail in another paper from Malawi (<https://www.medrxiv.org/content/10.1101/2023.08.22.23294324v1>).

Line 702: Presumably these samples were all positive on the rapid kit?

Response: Please see response 11 to reviewer 2 above.

Line 703: This line struck me. Why were samples only collected for patients with known antibiotic consumption?

Response: Unfortunately, we did not have this information. However, this is likely to be the case as over-the-counter antibiotic sales are common in Malawi and other African settings.

Line 745-763. Much as already been discussed about this. In recent studies we have been rooting the 7PET phylogeny on strain A6, and not including any pre-7PET or non7PET strains so you can actually get the resolution within the phylogenetic tree. If you don't kick those strains out, the 7PET just looks like a line and you can't see the branching patterns within the 7PET lineage.

Response: We agree that it's difficult to see the branching patterns of the 7PET strains without excluding the non-7PET strains. As suggested, we have rooted the 7PET phylogeny generated using Gubbins using A6 genome (GenBank assembly accession: GCF_001255575.1) as the outgroup. Indeed, visualising the 7PET strains separately improved the diversity within this lineage. For the phylogeny containing both 7PET and non-7PET strains, we rooted it using *V. mimicus* as the outgroup. We only used this tree to show the genetic relatedness of the *Vc* strains (7PET and non-7PET) from Malawi (Figure 1). However, we also generated another phylogeny containing only the global 7PET isolates after excluding recombinogenic regions using Gubbins and rooted using the A6 genome (GenBank assembly accession: GCF_001255575.1) as the outgroup (Figure 3). This phylogeny allowed us to see the branching patterns of the 7PET strains and to determine the potential geographical of the outbreak strains from Malawi.

I'm very much looking forward to the next iteration of this manuscript.

Response: Thank you for the insightful comments and for pointing out the issues with the whole-genome alignment. We hope we have addressed all the issues suggested.

Reviewer #3 (Remarks to the Author):

Review Report:

This study has a good potential for the understanding of the cholera epidemics in Malawi especially in the climate change context. However the work conducted here needs substantial improvements.

Response: Thank you for the excellent summary of the manuscript. We have addressed the specific comments and suggestions below.

Abstract: Lines 60-62 in the abstract do not provide clear understanding because these features described here are already known in the T10 and T13 7PET strains in the region. They could be new, but they don't seem so.

Response: Based on this and comments by the other reviewers, we repeated the analysis to include other recent Vc isolates from other countries. Our revised findings, consistent with the recent reports of imported Malawi strains to South Africa (PMID: 37352549) and another paper from Malawi (<https://www.medrxiv.org/content/10.1101/2023.08.22.23294324v1>), we have found that indeed the strains described in this manuscript represent a new introduction from Asia termed AFR15 or T15 rather T12 as previously stated.

Introduction: Lines 96-98 seem to be limited to T12 strains while ignoring recent literature that indicates the presence of T13 strains in East Africa, particularly in Tanzania, Kenya, and Uganda since 2017. Additionally, other T13 strains have been found in Zimbabwe in 2018, exhibiting newly acquired resistance profiles including an ESBL gene. Overall, the literature search needs to be more comprehensive without ignoring already known facts.

Response: Thank you for pointing out the literature regarding T12 and T13 strains in East Africa. We have now cited these papers in the introduction and other sections of the manuscript.

Virulence Factors: The ctxB genotype should not be regarded as a virulence factor but rather as a biotyping marker. **The focus should be on ctxA, which plays a crucial role in cholera virulence, as well as the TCPs and the other genes on the prophage as well as the VSPs. So far, no new findings have been reported regarding the virulence profile of these isolates, so this information could be summarized in a single sentence.** Furthermore, it should be considered that isolates lacking the ctxA/B complex may not be regarded as toxigenic, unless it is a sequencing issue, which should be discussed further. **Also, it would be valuable to confirm the presence of Ogawa and Inaba serotypes through genomic analysis of the wbeT gene.**

Response: We have revised the results paragraph describing the virulence factors. We have removed text focusing on other virulence factors outside the CTX, TCP, VPI, and

VSPs. We have described both *ctxB* and *ctxA* as they are present on the same prophage. Indeed, the strains lacking *ctxA/B* are toxigenic. As also suggested by reviewer #2, we have now performed further analysis of the mapping-based whole-genome alignment to confirm the presence of the CTX prophage that carry these genes.

We have now compared the *wbeT* gene of the Inaba and Ogawa *Vc* isolates from Malawi. We extracted the *wbeT* sequences (GenBank accession: JF284685) from the Inaba and Ogawa *Vc* isolates using BLASTN (PMID: 2231712) and “getfasta” option implemented in bedtools (PMID: 20110278). However, we did not find any specific mutations distinguishing these serotypes (Supplementary Data 3). These findings are consistent with findings from a study of serotype switching in Haiti which also reported that the serotype Inaba emergence was not linked to specific mutations in the *wbeT* gene (PMID: 36537825). We have updated the methods section and results section accordingly.

Global Phylogeny: The global phylogeny analysis is missing key guides to demonstrate the sublineages of these isolates. The main isolate of the Malawi outbreaks appear to belong to the T13 lineage. Considering the importance of this study and its implications for future reference in the global phylogeny of Malawi isolates, it is necessary to rerun the phylogenetic analysis. This analysis should include annotating clearly on the tree, all 7PET sublineages from T1 to T13, incorporating the latest T13 strains from Zimbabwe in 2018 (New England Journal) and East Africa (refer to EID and PLOS NTD papers from 2019/2023). Rooting the tree on the reference N16961 strain is essential. It is important for the authors to confirm whether the previous Malawi strains are T10 in their phylogeny analysis and report it accordingly to ensure universality in the field.

Response: We agree with this assessment. As similarly noted by reviewer #2, we have now repeated the phylogenetic analysis, including additional isolates from the suggested countries and the recent isolates from South Africa and Pakistan. Based on this revised phylogeny, we found that the 2022-2023 outbreak strains from Malawi are associated with the AFR15 or T15 lineage rather than T13, as we previously reported. We have revised the global 7PET phylogeny to show the sublineages from T1 to T15, as suggested (Figure 3). Regarding rooting the phylogeny using the N16961 genome, we would like to mention that we have now rooted the tree using the A6 genome (GenBank assembly accession: GCF_001255575.1) based on an earlier suggestion by reviewer #2. The A6 genome corresponds to the earliest sequenced 7PET isolate, which makes it suitable as an outgroup of the 7PET strains. Regarding the historical *Vc* isolates from Malawi, we can confirm that these strains belong to sublineages T1 and T5, as previously reported by Weill et al (PMID: 29123067), and we have mentioned this in the manuscript and cited the Weill et al paper.

Results and Discussion: Instances where the results are discussed or compared to other studies should be moved to the discussion section, as per the designated structure of the report.

Response: We have removed some discussion in the results where possible, where this may have been excessive. Although we agree with this suggestion, we would also like to mention that we find it generally useful to include a brief discussion or context of what the results mean immediately after describing the results, especially for genomics papers. This helps to keep the readers immediately understand what the findings mean before they read the full discussion, which generally keeps them engaged.

Environmental Isolate: The inclusion of a single environmental isolate is insufficient to draw conclusions regarding clinical versus environmental aspects.

Response: This is indeed correct. We have not focused much on the clinical versus environmental comparison. We have discussed this as one of the limitations of our study and have highlighted that this should be one of the focuses of future studies.

Resistance Profile: The resistance profile should be verified for comparison with other T10 and T13 strains, as this information is already known both for the genotypes and the MGEs. So these should be reported not as new findings but be put in existing context. None of these ICEs found are new.

Response: Based on the revised analysis, after including additional contextual genomes from Pakistan and other countries, we have found that our isolates belong to the newly reported AFR15 or T15 transmission lineage (PMID: 37352549), instead of T10 or T13 as previously reported. Since this is a new lineage, we think it would be great to report the resistance profile. As suggested, we have compared the MGEs in our isolates against isolates from other sublineages or genotypes found in the region that also carried these elements. Furthermore, we have also cited papers describing some of these MGEs, including those from African settings where lineages carrying these elements were also reported.

In conclusion it seems like the Malawi isolates are likely just a continuation of the existing sub lineages in the region, now a new introduction. Otherwise, authors need to argue for it more effectively with their data.

Response: Based on the revised analysis after including additional recent genomes from Pakistan and other countries, it seems that the Malawi isolates are associated with a new

introduction of the AFR15 or T15 sublineages rather than a continuation of the existing lineages in the region. These observations are consistent with those described in other recent papers from South Africa (PMID: 37352549) and Malawi (<https://www.medrxiv.org/content/10.1101/2023.08.22.23294324v1>).

Overall, to enhance the comprehensiveness of the report, it is necessary to address the mentioned points and incorporate the appropriate revisions and reorganizations.

Response: Thank you for excellently summarising the paper and insightful suggestions. We have addressed the suggested changes and revised the manuscript to cite literature that was omitted, highlighted that the new lineage represents a new introduction, and reorganised some text and sections to improve clarity as suggested.

REVIEWER COMMENTS

Reviewer #1 (Remarks to the Author):

All my previous comments have been adequately addressed. Overall, the revision process has made the paper stronger and much more convincing.

However, since the first round of reviews, a preprint has been published presenting similar data from Malawi (<https://www.medrxiv.org/content/10.1101/2023.08.22.23294324v1.full.pdf>). This preprint is incorrectly cited in the revised version of the manuscript (ref. 97). In addition, its content should be more closely compared with that of the current manuscript.

Reviewer #2 (Remarks to the Author):

The authors have addressed the major concern regarding the SNP alignment and now have produced results that are comparable with previous findings. They also addressed most of my other initial comments. However, now that the underlying analyses are correct, we need to get this manuscript sorted out as it still needs substantial work. This manuscript is an important contribution to the field and I want it to succeed, it just needs some attention to detail and tightening up of the text.

There is still too much redundancy, as other reviewers have also pointed out. There is much focus on comparison and contrasting historical strains – of which there are only six historical isolates. These six isolates were sequenced as part of the Weill et al 2017 paper (PMID 29123067) and characterized within that analysis. That analysis already showed that 4 of the 6 were part of T1, and the other two are T5. The supplementary table in that paper also fully characterizes these isolates, which includes AMR, virulence gene profile, CTX, wbeT, and SXT. Given that this current manuscript also demonstrates that the current outbreak is clearly not related to these historical samples, I do not understand the in-depth compare and contrast. I would urge the authors to consider removing much of these comparison analysis unless they can make a compelling argument why it is needed for us to understand the current outbreak strain.

This should be a rather simple story, and one that can be told more concisely and with much more clear organization. Everything you need to tell this story is in Figure 3. This clearly demonstrates that the recent outbreak is not related to previously circulating strains in Malawi, but is related to strains from South Asia and the Middle East, representing a novel introduction, which has previously been described as T15. There is apparently one T11 strain in 2022 as well (see notes below).

The epi analysis and the cholera attribution sections are good.

Just a general comment on the use of the MLST ST69. Nearly all 7PET isolates are ST69. Rarely, 7PET has a single-locus variant, ST515, in isolates from Africa belonging to lineage T10. The authors continuously refer to both ST69 and 7PET, which is the same thing and sometimes use the nomenclature “ST69 7PET isolate” which is redundant. The authors never describe in the paper what ST69 is referring to and begin its usage abruptly on line 245.

The next section describing the 7PET lineage in Malawi could very likely be combined with the next section on the overall larger phylogeny. Since you have placed all these isolates into the larger phylogeny, and this shows that the outbreak strain is, indeed, part of the 7PET lineage and part of T15 - all of the details about looking at the 7PET marker genes etc seem a bit redundant. The last paragraph in this section is essentially completely redundant when put into the context of the larger phylogeny in the next section.

Lines 232-252: I am now thoroughly confused by your finding of serotype Inaba, since this was all done in-silico and not via serology. The wild-type wbeT gene confers the Ogawa serotype.

Disruptions or mutations in this gene confer the Inaba serotype. Since you were doing this analysis via sequence comparison, this by definition had to be based on differences between the wbeT in Ogawa and Inaba strains. The papers referred to by the authors have reported the puzzling findings of serologically typed Inaba strains without finding mutations in the wbeT gene. Here, you are only looking at sequence data and do not have the serology. The only way I can think that you are getting Inaba without finding mutations in wbeT, is if you placed some of these Inaba wbeT genes in your database from the above papers which don't have mutations. I would recommend looking at the actual read mapping data in those you have called Inaba. If you do not find mutations, these strains should be typed Ogawa in the absence of serological data to say otherwise.

Line 327: The 2015 Zimbabwe isolates are T11, not T12. These were sequenced as part of the NEJM paper from Mashe et al. The supplementary tables fully describe these strains. The T11 isolates harbor ctxB1, which your 2022 strain appears to also share – corroborating its placement. The resistance profiles also match that of the T11 strain.

I would organize the description of the non-7PET isolates as a separate small section, and remove the intermingling of these results with the outbreak strain. This would help simplify the overall paper.

The virulence gene section reports that there is nothing of note that distinguishes this current outbreak strain from other concurrently circulating strains over the past decade, but yet it takes up a full page of text – mostly due to the comparison to the six historical strains.

The section of AMR and the SXT could benefit the most from shortening. Perhaps some of this text, like lines 438-447 should be moved to the methods section.

Lines 387-391: Six lines of text simply to list out all the antibiotics that are screened by PathogenWatch – I would move this to the methods section.

The ciprofloxacin resistance is conferred via point mutations in gyrA and parC, however this not mentioned in the text, nor is it listed in the supplementary tables. Please update.

Line 410: I'm guessing the one isolate that didn't match was the single T11 isolate? This could easily be clarified by stating "All T15 isolates from Malawi harbored..., while the single T11 isolate harbored...."

Line 426: Typo as you actually detected catB9 not catB7 (as per your supplement data) catB9 is widely known to NOT confer chloramphenicol resistance in VC (<https://doi.org/10.1046/j.1365-2958.2002.02861.x>) .

Line 428: no need to list all tet genes, just say we did not find tetracycline resistance genes.

Line 435: same thing, no need to list all the bla genes here if you did not find any.

Line 470: Refer back to my comments about the Inaba serotype switching. I would be careful about stating this without serological data, given you did not find mutations in wbeT.

Line 486: Typo "Africa23 . U"

Lines 500-501: Perhaps a rephrasing. "These findings suggests that it is possible that profuse diarrhea cases can be incorrectly attributed to Vc."

Line 549-552: You have not provided data to back up this claim that the VSP-II and ctxB7 explained higher transmission and virulence.

Line 62. "climatic changes" not the right choice. This wasn't a manuscript where you linked say, a 2C global temp rise with increased cholera. This was more a direct result of actual weather events,

in this case, cyclones.

Line 123 : Awkwardly written sentence.

Line 137: Smith et al EID paper would more correctly be the first insight.

Don't start talking about Malawi until 5th paragraph in the Introduction.

Line 131: strains rather than clones.

Line 146: write as of the actual date from which the numbers are being reported.

Line 193: I would end the sentence right after Malawi and cut the rest.

Line 232: lower-case "we"

Line 391: furazolidone listed twice

I'm not sure there is even a need for Figure 2. This figure is essentially the same set of strains represented 3 times. The actual 7PET outbreak strains are all in Figure 3, which makes the point much more clearly than in Fig 2.

Figure 3:

1. Please ensure column color schemes are consistent between the two panels. For instance the serogroup columns are differently colored between the two.

2. It seems the tree nodes are colored based of "ST69" and then there is also a column for ST (which is all one color). The nodes in the tree could be colored by something much more meaningful, such as location or strains in this study. I would remove the ST column as it provides no information.

Figure 4:

1. Again here, the use of ST and coloring nodes by ST69 provides no additional information. Consider coloring nodes by something meaningful.

2. The country column here is not needed as they are all from Malawi.

3. For the AMR, I would much rather see the actual genes P/A rather than this higher-level output from PathogenWatch listing the antibiotic names.

Reviewer #3 (Remarks to the Author):

Dear authors and editor,

I have reviewed the revised manuscript, figures, and the authors' response to reviewer questions. The improvements made since the initial version are outstanding, and I am generally satisfied with this revised version. I have a few general comments that I would like to bring to your attention.

1. Reference to Preprint Online:

- The reference to the preprint online supporting the AFR15 introduction is noted. However, it is advisable to carefully consider the validity of the AFR15 argument, as preprints are not peer-reviewed articles. But yes these are two studies of the same epidemics and they both seem to point to the same findings..

2. Climate Change and Epidemic Context:

- The link to climate change, population displacement, and sudden deterioration in hygiene

conditions as factors favoring the epidemic is clear in this epidemic. However, given the context leading to the epidemic, where a persistent sublineage already existing in the region is expected, the relevance of attributing the epidemic to a new introduction seem unexpected and could be added to discussion. And as this is a new introduction, I'm wondering if the climate change reference in this context is still relevant.

3. Pangenomic Display:

- Given the similarity in MGE content between AFR15 and the sublineage AFR10 predominantly circulating in the region, it is suggested that a pangenomic display illustrating the genomic differences between these two sublineages be included. This could be presented as supplementary material to enhance the comprehensiveness of the analysis.

4. Distinguishing from the Preprint:

- Both the authors and the editor are encouraged to consider how to distinguish this study from the existing preprint, which also describes the same epidemic at a similar genomic level. Emphasizing what is new or complementary in this study compared to the preprint will enhance the value for the readership. Instead of focusing too much on the new introduction, consideration could be given to highlighting genomic differences from other circulating sublineages and exploring potential environmental influences leading to these differences.

5. Emergence of New Sublineages:

- Considering the emerging pattern of cholera as a disease of Africa and the suspicion that what may be perceived as new introductions from Asia could be local emergences, it is suggested to discuss this perspective. The manuscript could explore the possibility that new sublineages may be a result of changes in local strains. While acknowledging the limitation of environmental samples in supporting the emergence and spread of new sublineages following climate disasters, addressing this aspect would contribute to a more comprehensive discussion.

Overall, the manuscript has made significant progress, and addressing these points will further enhance its clarity and contribution. Thank you for considering these comments, and I look forward to the final version of the manuscript.

Sincerely,

REVIEWER COMMENTS

Reviewer #1 (Remarks to the Author):

All my previous comments have been adequately addressed. Overall, the revision process has made the paper stronger and much more convincing.

However, since the first round of reviews, a preprint has been published presenting similar data from Malawi (<https://www.medrxiv.org/content/10.1101/2023.08.22.23294324v1.full.pdf>). This preprint is incorrectly cited in the revised version of the manuscript (ref. 97). In addition, its content should be more closely compared with that of the current manuscript.

Response: Thank you for your excellent comments and suggestions. We have checked the citation of the aforementioned preprint (now reference #91) and have included a discussion of the additional analyses described in the preprint. We have mentioned in the discussion that this preprint describes phylogeographic analyses of the strains using BEAST (PMID: 17996036), which complements our findings which were based on maximum likelihood phylogenetic analysis.

Reviewer #2 (Remarks to the Author):

The authors have addressed the major concern regarding the SNP alignment and now have produced results that are comparable with previous findings. They also addressed most of my other initial comments. However, now that the underlying analyses are correct, we need to get this manuscript sorted out as it still needs substantial work. This manuscript is an important contribution to the field and I want it to succeed, it just needs some attention to detail and tightening up of the text.

Response: Thank you for the positive feedback. We have addressed your specific comments below.

There is still too much redundancy, as other reviewers have also pointed out. There is much focus on comparison and contrasting historical strains – of which there are only six historical isolates. These six isolates were sequenced as part of the Weill et al 2017 paper (PMID 29123067) and characterized within that analysis. That analysis already showed that 4 of the 6 were part of T1, and the other two are T5. The supplementary table in that paper also fully characterizes these isolates, which includes AMR, virulence gene profile, CTX, wbeT, and SXT. Given that this current manuscript also demonstrates that the current outbreak is clearly not related to these historical samples, I do not understand the in-depth compare and contrast. I would urge the authors to consider removing much of these comparison analysis unless they can make a compelling argument why it is needed for us to understand the current outbreak strain.

This should be a rather simple story, and one that can be told more concisely and with much more clear organization. Everything you need to tell this story is in Figure 3. This clearly demonstrates that

the recent outbreak is not related to previously circulating strains in Malawi, but is related to strains from South Asia and the Middle East, representing a novel introduction, which has previously been described as T15. There is apparently one T11 strain in 2022 as well (see notes below).

Response: We have revised the manuscript to remove most of the text related to the comparison of the Malawi 2022-2023 isolates to the historical isolates. However, we have included a brief description of the current outbreak strains in relation to the historical isolates for context.

The epi analysis and the cholera attribution sections are good.

Response: Thank you for this positive feedback.

Just a general comment on the use of the MLST ST69. Nearly all 7PET isolates are ST69. Rarely, 7PET has a single-locus variant, ST515, in isolates from Africa belonging to lineage T10. The authors continuously refer to both ST69 and 7PET, which is the same thing and sometimes use the nomenclature “ST69 7PET isolate” which is redundant. The authors never describe in the paper what ST69 is referring to and begin its usage abruptly on line 245.

Response: We have removed the redundant use of the MLST ST69 in favour of the 7PET terminology. We have mentioned in the results section describing the MLST clones that ST69 refers to the 7PET strains and then used the 7PET term throughout the manuscript.

The next section describing the 7PET lineage in Malawi could very likely be combined with the next section on the overall larger phylogeny. Since you have placed all these isolates into the larger phylogeny, and this shows that the outbreak strain is, indeed, part of the 7PET lineage and part of T15 - all of the details about looking at the 7PET marker genes etc seem a bit redundant. The last paragraph in this section is essentially completely redundant when put into the context of the larger phylogeny in the next section.

Response: We agree that the mentioned paragraphs contained redundant details. To address this, we have removed three paragraphs from the “*The 2022–2023 outbreak in Malawi was primarily driven by the 7PET lineage O1 Ogawa serotype*” section which described similar results to the next section titled “*Genetic analysis suggests the 2022–2023 outbreak-associated V. cholera isolates were recently imported from Asia*”. However, we have maintained the two sections as separate as the first section focuses only on the Malawi strains and the next one places the Malawi strains in the global context. We would like to describe the local context first before the global context. Also, we believe this would flow better based on how we have designed the figures, i.e., Fig. 2, showing the phylogeny of the Malawi strains, and Fig. 3 depicting the Malawi strains in the global context.

Lines 232-252: I am now thoroughly confused by your finding of serotype Inaba, since this was all done in-silico and not via serology. The wild-type wbeT gene confers the Ogawa serotype. Disruptions or mutations in this gene confer the Inaba serotype. Since you were doing this analysis via sequence

comparison, this by definition had to be based on differences between the wbeT in Ogawa and Inaba strains. The papers referred to by the authors have reported the puzzling findings of serologically typed Inaba strains without finding mutations in the wbeT gene. Here, you are only looking at sequence data and do not have the serology. The only way I can think that you are getting Inaba without finding mutations in wbeT, is if you placed some of these Inaba wbeT genes in your database from the above papers which don't have mutations. I would recommend looking at the actual read mapping data in those you have called Inaba. If you do not find mutations, these strains should be typed Ogawa in the absence of serological data to say otherwise.

Response: We have revised the text and figures to mention that all the 7PET isolates from the 2022–2023 cholera outbreak in Malawi belonged to the serogroup O1 Ogawa serotype.

Line 327: The 2015 Zimbabwe isolates are T11, not T12. These were sequenced as part of the NEJM paper from Mashe et al. The supplementary tables fully describe these strains. The T11 isolates harbor ctxB1, which your 2022 strain appears to also share – corroborating it's placement. The resistance profiles also match that of the T11 strain.

Response: We have updated the sentence to indicate that the 2015 Zimbabwe isolates belonged to T11, not T12 lineage.

I would organize the description of the non-7PET isolates as a separate small section, and remove the intermingling of these results with the outbreak strain. This would help simplify the overall paper.

Response: Based on the comment above regarding redundant details, we have removed most of the text on non-7PET isolates after removing three paragraphs in the “*The 2022–2023 outbreak in Malawi was primarily driven by the 7PET lineage O1 Ogawa serotype*” section. Therefore, the current version of the manuscript only briefly describes these non-7PET isolates. We believe that the current description of the non-7PET isolates is minimal and does not detract from the results for the outbreak strains.

The virulence gene section reports that there is nothing of note that distinguishes this current outbreak strain from other concurrently circulating strains over the past decade, but yet it takes up a full page of text – mostly due to the comparison to the six historical strains.

Response: We agree that the description of the virulence genes was too long and somewhat unnecessary. We have cut down this section to a single paragraph to state that the 2022–2023 Malawi 7PET isolates harboured the same set of virulence factors typically found in 7PET strains, as previously described by others.

The section of AMR and the SXT could benefit the most from shortening. Perhaps some of this text, like lines 438-447 should be moved to the methods section.

Response: We have trimmed down the section on AMR and the SXT element by nearly half the original length. We have removed the text previously in lines 438-447 as we have already described it in the methods section.

Lines 387-391: Six lines of text simply to list out all the antibiotics that are screened by PathogenWatch – I would move this to the methods section.

Response: We have moved the description of the antibiotics to the methods section.

The ciprofloxacin resistance is conferred via point mutations in *gyrA* and *parC*, however this is not mentioned in the text, nor is it listed in the supplementary tables. Please update.

Response: We have revised the manuscript as follows: “*We also observed intermediate resistance against ciprofloxacin (CIP), which is an alternative treatment option in children in Malawi, conferred by point mutations in the *gyrA* and *parC* genes⁸¹.*” We have also updated supplementary Data 2 to include data on the SNPs associated with ciprofloxacin resistance in the isolates.

Line 410: I’m guessing the one isolate that didn’t match was the single T11 isolate? This could easily be clarified by stating “All T15 isolates from Malawi harbored...., while the single T11 isolate harbored....”

Response: The one isolate that lacked the *aph(3'')*-Ib and *aph(6)*-Id genes previously described in line 410 was T15 not T11. We have revised the sentence as follows: “Nearly all the Malawi 2022–2023 outbreak-associated 7PET isolates sequenced in this study harboured *aph(3'')*-Ib and *aph(6)*-Id, and the dual presence of *strA* and *strB*, which contributed to the streptomycin (STR) resistance”.

Line 426: Typo as you actually detected catB9 not catB7 (as per your supplement data) catB9 is widely known to NOT confer chloramphenicol resistance in VC (<https://doi.org/10.1046/j.1365-2958.2002.02861.x>).

Response: Thank you for noticing this. We have corrected the typo.

Line 428: no need to list all tet genes, just say we did not find tetracycline resistance genes.
Line 435: same thing, no need to list all the bla genes here if you did not find any.

Response: We have revised the sentences to remove all the tetracycline and carbapenem resistance genes

Line 470: Refer back to my comments about the Inaba serotype switching. I would be careful about stating this without serological data, given you did not find mutations in *wbeT*.

Response: As suggested above, we have revised the serotyping results to state that all the 2022–2023 outbreak strains were associated with the O1 Ogawa serotype.

Line 486: Typo “Africa23 . U”

Response: We have corrected the typo.

Lines 500-501: Perhaps a rephrasing. “These findings suggest that it is possible that profuse diarrhea cases can be incorrectly attributed to Vc.”

Response: We have rephrased the sentence as suggested.

Line 549-552: You have not provided data to back up this claim that the VSP-II and *ctxB7* explained higher transmission and virulence.

Response: We agree with this assessment. We have revised the sentence to mention that the VSP-II and *ctxB7* explained the higher transmission and virulence of the 2022–2023 outbreak strains as follows: “*The major distinguishing genetic characteristic of the 2022–2023 isolates from the historical isolates is the presence of a different version of the VSP-II pathogenicity island and ctxB7 genotype, although it’s unlikely that these differences may partly explain the observed transmission and virulence of the 2022–2023 O1 strains in Malawi.*”

Line 62. “climatic changes” not the right choice. This wasn’t a manuscript where you linked say, a 2C global temp rise with increased cholera. This was more a direct result of actual weather events, in this case, cyclones.

Response: We have replaced “climatic changes” with “devastating cyclones”.

Line 123 : Awkwardly written sentence.

Response: We have revised the sentence as follows: “Besides the devastation caused in Malawi by tropical cyclones Ana and Gombe in early 2022 and Freddy in 2023, specific attributes of the Vc strains that may have contributed to the high incidence and mortality of the 2022–2023 cholera outbreak remain unknown.”

Line 137: Smith et al EID paper would more correctly be the first insight.

Response: We posted our preprint on medrxiv on June 12, 2023 before the Smith et al EID paper was published on June 23, 2023. However, we have referenced the Smith et al paper throughout the manuscript.

Don't start talking about Malawi until 5th paragraph in the Introduction.

Response: We have combined paragraphs 4 and 5 and revised the text to mention Malawi in the last paragraph of the introduction as suggested.

Line 131: strains rather than clones.

Response: We have replaced "clones" with "strains".

Line 146: write as of the actual date from which the numbers are being reported.

Response: We have mentioned the total number of cases and deaths from 2022 to now.

Line 193: I would end the sentence right after Malawi and cut the rest.

Response: We have revised the sentence as suggested.

Line 232: lower-case "we"

Response: We have written "we" in lower case.

Line 391: furazolidone listed twice

Response: We have deleted the repeated word.

I'm not sure there is even a need for Figure 2. This figure is essentially the same set of strains represented 3 times. The actual 7PET outbreak strains are all in Figure 3, which makes the point much more clearly than in Fig 2.

Response: Although we understand this suggestion, we would prefer to have both figure 2 and 3 in the manuscript. As mentioned in our previous responses to the comments, we would like to include Fig. 2 as it shows only the Malawi isolates. Most importantly, Fig. 2 shows all the Malawi isolates from both 7PET and non-7PET lineages. In contrast, although similar information is shown in Fig. 3, the phylogeny only contains the outbreak, i.e., 7PET strains, as we were asked to do when responding to the previous reviewers' comments. Considering we have described the results from local to global context, we would like to maintain the same structure by having both Fig. 2 and 3 in the manuscript.

Figure 3:

1. Please ensure column color schemes are consistent between the two panels. For instance the serogroup columns are differently colored between the two.

2. It seems the tree nodes are colored based of “ST69” and then there is also a column for ST (which is all one color). The nodes in the tree could be colored by something much more meaningful, such as location or strains in this study. I would remove the ST column as it provides no information.

Response: We have revised the colour scheme of Fig. 3 to make the colours consistent in the two panels. We have also coloured the nodes in the tree based on location of the strains and removed the ST column.

Figure 4:

1. Again here, the use of ST and coloring nodes by ST69 provides no additional information. Consider coloring nodes by something meaningful.

2. The country column here is not needed as they are all from Malawi.

3. For the AMR, I would much rather see the actual genes P/A rather than this higher-level output from PathogenWatch listing the antibiotic names.

Response: We removed ST column and coloured the nodes by isolation period of the strains. Although we understand the comment regarding showing gene presence and absence instead of the antibiotic susceptibility profiles, we prefer to show the antibiotic susceptibility profiles as this is easier to understand by the general reader and because there would be too many resistance genes to show in the figure. Therefore, we have included detailed information on the actual gene presence and absence in the supplementary material.

Reviewer #3 (Remarks to the Author):

Dear authors and editor,

I have reviewed the revised manuscript, figures, and the authors' response to reviewer questions. The improvements made since the initial version are outstanding, and I am generally satisfied with this revised version. I have a few general comments that I would like to bring to your attention.

Response: Thank you for your feedback and excellent summary of the manuscript. We have addressed your additional comments below.

1. Reference to Preprint Online:

- The reference to the preprint online supporting the AFR15 introduction is noted. However, it is advisable to carefully consider the validity of the AFR15 argument, as preprints are not peer-reviewed articles. But yes these are two studies of the same epidemics and they both seem to point to the same findings..

Response: We agree that preprints are not peer-reviewed articles. However, we have provided two references, one to a preprint and another to a peer-reviewed paper published in the Emerging Infectious Diseases journal (PMID: 37352549), which described the AFR15 introduction.

2. Climate Change and Epidemic Context:

- The link to climate change, population displacement, and sudden deterioration in hygiene conditions as factors favoring the epidemic is clear in this epidemic. However, given the context leading to the epidemic, where a persistent sublineage already existing in the region is expected, the relevance of attributing the epidemic to a new introduction seems unexpected and could be added to discussion. And as this is a new introduction, I'm wondering if the climate change reference in this context is still relevant.

Response: Based on the genomic analysis of the 2022–2023 Malawi strains in the context of strains in the region and globally, the evidence supports a new introduction from Asia. Therefore, we believe that it's the combination of a new introduction and most importantly the climatic events, i.e., cyclones that led to the cholera epidemic in Malawi. Based on this and previous reviewers' comments, we have revised the manuscript to mention that "The combination of the devastating cyclones and introduction of a new Vc strain in Malawi offered a perfect opportunity for the outbreak."

3. Pangenomic Display:

- Given the similarity in MGE content between AFR15 and the sublineage AFR10 predominantly circulating in the region, it is suggested that a pangenomic display illustrating the genomic differences between these two sublineages be included. This could be presented as supplementary material to enhance the comprehensiveness of the analysis.

Response: We have included a new sheet "Virulence genes all (ABRicate)" to Supplementary Data 2 to show the gene virulence presence and absence in the Malawi strains (AFR15) and all the contextual strains from elsewhere included in the analysis (which included the AFR10 and other lineages).

4. Distinguishing from the Preprint:

- Both the authors and the editor are encouraged to consider how to distinguish this study from the existing preprint, which also describes the same epidemic at a similar genomic level. Emphasizing what is new or complementary in this study compared to the preprint will enhance the value for the readership. Instead of focusing too much on the new introduction, consideration could be given to highlighting genomic differences from other circulating sublineages and exploring potential environmental influences leading to these differences.

Response: The main difference between our study and the preprint is the phylogeographic analysis using BEAST. Since our manuscript was posted on medrxiv before the other preprint came out, we performed more extensive analyses on disease incidence and genomic analyses. We have already mentioned that our manuscript does not perform the phylogeographic analyses

and that these findings are provided in the Chabuka et al preprint (<https://www.medrxiv.org/content/10.1101/2023.08.22.23294324v1>).

5. Emergence of New Sublineages:

- Considering the emerging pattern of cholera as a disease of Africa and the suspicion that what may be perceived as new introductions from Asia could be local emergences, it is suggested to discuss this perspective. The manuscript could explore the possibility that new sublineages may be a result of changes in local strains. While acknowledging the limitation of environmental samples in supporting the emergence and spread of new sublineages following climate disasters, addressing this aspect would contribute to a more comprehensive discussion.

Response: We have added additional text in the discussion to highlight the possibility that the new sublineages may be a result of changes in local strains as follows: "While there is a possibility that the Malawi 2022–2023 outbreak strain may have circulated locally and in the region, this is difficult to prove decisively due to the limited availability of genomic data from Africa."

Overall, the manuscript has made significant progress, and addressing these points will further enhance its clarity and contribution. Thank you for considering these comments, and I look forward to the final version of the manuscript.

Response: Thank you for the excellent feedback.

REVIEWERS' COMMENTS

Reviewer #2 (Remarks to the Author):

Well done to the authors! I appreciate their effort and persistence on this manuscript. I am very self-aware that I was coming off as a "persnickety" reviewer, but I think the final manuscript here is significantly better than the earlier versions. The authors should be pleased with the result, as it represents much time and effort put into it. I will be very happy to see it published!

I have very minimal comments on this final version.

Line 329: the catB9 gene is known to NOT confer chloramphenicol resistance in VC. Therefore, these strains are likely not resistant to CHL based on the presence of this gene.

See this paper for reference: <https://onlinelibrary.wiley.com/doi/full/10.1046/j.1365-2958.2002.02861.x>)

Line 333-336: I'm not sure that varG has been directly implicated to confer carbapenem resistance in VC. The paper you cite utilizes E coli for lab experiments by cloning the VC var operon. That paper also demonstrate moderate activity of VarG in E coli against cefepime in addition to carbapenems. Carbapenem resistance in VC is still, thankfully, very limited and almost always mediated by the acquisition of a known resistance gene, like blaNDM-1.

<https://www.pnas.org/doi/full/10.1073/pnas.1900141116>. The same goes for ESBL related phenotypes. As varG seems conserved in VC, the finding that most strains are not intrinsically resistant to 3rd generation cephalosporins, again warrants tempered statements about the in-vivo effect and resistance phenotypes conferred by varG. Without phenotypic data to back it up, I would be very cautious about stating all these strains are carbapenem resistant based on presence of varG.

Figure 2: You write "The circles with different colours at the tip of the phylogeny represent the year of isolation." However, the nodes aren't colored by year, they are still colored by Sequence Type. For all panels, sequence type has two legends, one for coloring the nodes and again for the color strip. In A, the colors do not match between the ST color strip and the ST colors for the nodes.

As these are minor points above, if the authors address them, I recommend publication without the need to see a revised manuscript again.

Reviewer #2 (Remarks to the Author):

Well done to the authors! I appreciate their effort and persistence on this manuscript. I am very self-aware that I was coming off as a “persnickety” reviewer, but I think the final manuscript here is significantly better than the earlier versions. The authors should be pleased with the result, as it represents much time and effort put into it. I will be very happy to see it published!

I have very minimal comments on this final version.

Response: Thank you for your comments, which greatly improved the manuscript.

Line 329: the *catB9* gene is known to NOT confer chloramphenicol resistance in VC. Therefore, these strains are likely not resistant to CHL based on the presence of this gene.

See this paper for reference: <https://onlinelibrary.wiley.com/doi/full/10.1046/j.1365-2958.2002.02861.x>)

Response: We have revised the sentence to mention that the *catB9* gene does not confer chloramphenicol resistance in *Vibrio cholerae*.

Line 333-336: I'm not sure that *varG* has been directly implicated to confer carbapenem resistance in VC. The paper you cite utilizes *E coli* for lab experiments by cloning the VC *var* operon. That paper also demonstrate moderate activity of *VarG* in *E coli* against cefepime in addition to carbapenems. Carbapenem resistance in VC is still, thankfully, very limited and almost always mediated by the acquisition of a known resistance gene, like *blaNDM-1*. <https://www.pnas.org/doi/full/10.1073/pnas.1900141116>. The same goes for ESBL related phenotypes. As *varG* seems conserved in VC, the finding that most strains are not intrinsically resistant to 3rd generation cephalosporins, again warrants tempered statements about the in-vivo effect and resistance phenotypes conferred by *varG*. Without phenotypic data to back it up, I would be very cautious about stating all these strains are carbapenem resistant based on presence of *varG*.

Response: We have revised the sentences regarding carbapenem resistance to temper down the assertion that resistance is based on the presence of the *varG* gene. We have also mentioned that phenotypic data is required to validate the phenotypic resistance profiles described in this study.

Figure 2: You write “The circles with different colours at the tip of the phylogeny represent the year of isolation.” However, the nodes aren't colored by year, they are still colored by Sequence Type. For all panels, sequence type has two legends, one for coloring the nodes and again for the color strip. In A, the colors do not match between the ST color strip and the ST colors for the nodes.

Response: We have revised Figure 2 to colour the nodes at the tips of the phylogeny by year of isolation.

As these are minor points above, if the authors address them, I recommend publication without the need to see a revised manuscript again.

Response: Thank you for your comments.